# BayeSQP: Bayesian Optimization through Sequential Quadratic Programming

**Paul Brunzema    Sebastian Trimpe**
Institute for Data Science in Mechanical Engineering
RWTH Aachen University
Aachen, Germany
`{brunzema, trimpe}@dsme.rwth-aachen.de`

## Abstract

We introduce `BayeSQP`, a novel algorithm for general black-box optimization that merges the structure of sequential quadratic programming with concepts from Bayesian optimization. `BayeSQP` employs second-order Gaussian process surrogates for both the objective and constraints to jointly model the function values, gradients, and Hessian from only zero-order information. At each iteration, a local subproblem is constructed using the GP posterior estimates and solved to obtain a search direction. Crucially, the formulation of the subproblem explicitly incorporates uncertainty in both the function and derivative estimates, resulting in a tractable second-order cone program for high probability improvements under model uncertainty. A subsequent one-dimensional line search via constrained Thompson sampling selects the next evaluation point. Empirical results show that `BayeSQP` outperforms state-of-the-art methods in specific high-dimensional settings. Our algorithm offers a principled and flexible framework that bridges classical optimization techniques with modern approaches to black-box optimization.

## 1   Introduction

In recent years, Bayesian optimization (BO) has emerged as a powerful framework for black-box optimization ranging from applications in robotics [9, 6, 44] to hyperparameter tuning [55, 12] and drug discovery [21, 40, 10]. To address high-dimensional problems emerging in these fields, a variety of high-dimensional BO (HDBO) approaches have been proposed, including the use of local BO (LBO) methods [15, 43] or methods that exploit specific structure in the objective [13]. Recently, a growing debate has emerged over whether such HDBO methods are truly necessary, given that appropriate scaling of the prior can already yield strong performance on certain high-dimensional benchmarks [30, 67]. However, as shown by Papenmeier et al. [48], these approaches solve numerical issues in the hyperparameter optimization of the Gaussian process (GP) surrogate but their success can still be attributed to emerging local search behavior. We argue that it is not a matter of choosing either HDBO approaches or standard approaches, but rather of leveraging recent advances in how to achieve numerical stability also for HDBO methods.

Building on this perspective, we aim to integrate the strengths of established classical optimization techniques within the HDBO framework. Specifically, we extend the widely-adopted local method for HDBO `GIBO` [43, 45, 65, 16, 23]—which can be interpreted as combining BO with first-order optimization methods—to LBO with second-order methods. We introduce `BayeSQP`, a novel algorithm for black-box optimization that merges the structure of sequential quadratic programming (SQP) with concepts from BO. `BayeSQP` employs GP surrogates for both the objective and constraints that jointly model the function values, gradients, and Hessians from only zero-order information (Figure 1). At each iteration, a local subproblem is constructed using the GP posterior estimates and solved to

39th Conference on Neural Information Processing Systems (NeurIPS 2025).

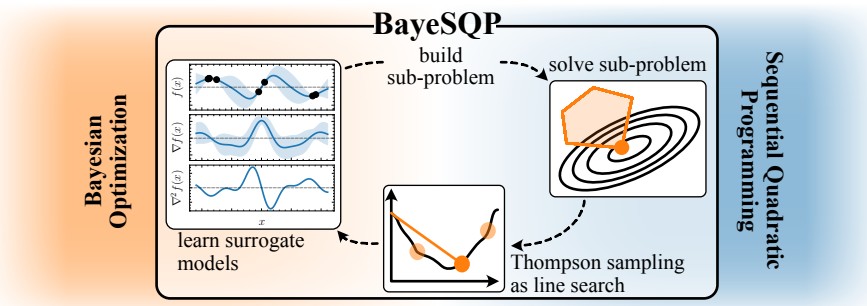

Figure 1: *Overview.* BayeSQP combines ideas from sequential quadratic programming and Bayesian optimization for efficient high-dimensional black-box optimization.

obtain a search direction. Through constrained Thompson sampling, we select the point for the next iteration. In summary, the key contributions of this paper are:

**C1** A novel algorithm BayeSQP leveraging GP surrogates to utilize the structure of classic SQP within BO for efficient high-dimensional black-box optimization with constraints.

**C2** An uncertainty-aware subproblem for BayeSQP that accounts for the variance and covariance in function and gradient estimates, resulting in a tractable second-order cone program.

**C3** Empirical experiments demonstrating that BayeSQP outperforms state-of-the-art BO methods in specific high-dimensional constrained settings.

## 2 Problem formulation

We consider the problem of finding an optimizer to the general non-convex optimization problem

$$\boldsymbol{x}^* = \arg\min_{\boldsymbol{x}\in\mathbb{X}} f(\boldsymbol{x}) \quad \text{subject to} \quad c_i(\boldsymbol{x}) \geq 0, \quad \forall i \in \mathbb{I}_m \coloneqq \{1,\dots,m\} \tag{1}$$

where $f : \mathbb{X} \to \mathbb{R}$ and constraints $c_i : \mathbb{X} \to \mathbb{R}$ for all $i \in \mathbb{I}_m$ are black-box functions defined over the compact set $\mathbb{X} \subset \mathbb{R}^d$. At each iteration $t \in \mathbb{I}_T$ where $T$ is the total budget for the optimization, an algorithm selects a query point $\boldsymbol{x}_t \in \mathbb{X}$ and receives noisy zeroth-order feedback following the standard observation model in BO as $f_t = f(\boldsymbol{x}_t) + \varepsilon_f$ for the objective, and $c_{i,t} = c_i(\boldsymbol{x}_t) + \varepsilon_{c_i}$ for all $i \in \mathbb{I}_m$ for the constraints where $\varepsilon_f$ and $\varepsilon_{c_i}$ are independent realizations from a zero-mean Gaussian distribution with possibly different noise variances. From these observations, we construct independent datasets for the objective function $\mathcal{D}_f^t = \{(\boldsymbol{x}_j, f_j)\}_{j=1}^t$ and for each constraint $\mathcal{D}_{c_i}^t = \{(\boldsymbol{x}_j, c_{i,j})\}_{j=1}^t$ for all $i \in \mathbb{I}_m$, which any zero-order method can leverage to solve (1).

## 3 Preliminaries

### 3.1 Sequential quadratic programming

SQP represents a powerful framework for solving nonlinear constrained optimization problems by iteratively solving quadratic subproblems. This method has become one of the most effective techniques for handling a wide range of optimization problems. The foundation of constrained optimization rests on the Lagrangian function, defined as $\mathcal{L}(\boldsymbol{x},\boldsymbol{\xi}) = f(\boldsymbol{x}) - \sum_{i=1}^m \xi_i c_i(\boldsymbol{x})$. It combines the objective with the constraints, where each constraint is weighted by its Lagrange multiplier $\xi_i$. Solving the optimization problem involves satisfying the Karush-Kuhn-Tucker conditions for this Lagrangian. To achieve this, at each iteration $t$, SQP constructs a quadratic approximation of the Lagrangian using the Hessian $\boldsymbol{H}_t = \nabla^2_{\boldsymbol{xx}}\mathcal{L}(\boldsymbol{x}_t,\boldsymbol{\xi})$ (or an appropriate approximation thereof) and linearizes the constraints around the current point $\boldsymbol{x}_t$. This generates the following subproblem:

$$\boldsymbol{p}_t = \arg\min_{\boldsymbol{p}\in\mathbb{R}^d} \quad \frac{1}{2}\boldsymbol{p}^\top \boldsymbol{H}_t \boldsymbol{p} + \nabla f(\boldsymbol{x}_t)^\top \boldsymbol{p} + f(\boldsymbol{x}_t)$$
$$\text{subject to} \quad \nabla c_i(\boldsymbol{x}_t)^\top \boldsymbol{p} \geq -c_i(\boldsymbol{x}_t), \ \forall i \in \mathbb{I}_m \tag{2}$$

The solution $\boldsymbol{p}_t$ provides a search direction, and the next iterate is typically computed as $\boldsymbol{x}_{t+1} = \boldsymbol{x}_t + \alpha_t \boldsymbol{p}_t$, where $\alpha_t$ is a step size determined by an appropriate line search procedure that ensures adequate progress toward the optimum. For an overview of classical SQP methods, see [46].

Under standard assumptions, SQP exhibits local superlinear convergence when using exact Hessian information, and various quasi-Newton approximation schemes (such as BFGS or SR1 updates, cf. [46]) can maintain good convergence properties while reducing computational overhead. This fast local convergence also makes it interesting for HDBO. However, the key challenge here is that usually only zero-order information on the objective and constraints is available.

## 3.2 Gaussian processes

GPs are a powerful and flexible framework for modeling functions in a non-parametric way. A GP is defined as a collection of random variables, any finite number of which have a joint Gaussian distribution. Formally, a GP is defined by its mean function $m(\boldsymbol{x}) := \mathbb{E}[f(\boldsymbol{x})]$ and kernel $k(\boldsymbol{x}, \boldsymbol{x}') := \mathrm{Cov}[f(\boldsymbol{x}), f(\boldsymbol{x}')]$ [53]. In BayeSQP, we use GPs to model the objective function $f$ and the constraints $c_i$ as standard in BO [18]. Contrary to standard BO, we aim to leverage the following property of GPs: They are closed under linear operations, i.e., the derivative of a GP is again a GP given that the kernel is sufficiently smooth [53]. This enables us to derive a distribution for the gradient and Hessian. We can formulate the following joint prior distribution:

$$\begin{bmatrix} \boldsymbol{y} \\ f \\ \nabla f \\ \nabla^2 f \end{bmatrix} \sim \mathcal{N}\left( \begin{bmatrix} m(\boldsymbol{X}) \\ m(\boldsymbol{x}) \\ \nabla m(\boldsymbol{x}) \\ \nabla^2 m(\boldsymbol{x}) \end{bmatrix}, \begin{bmatrix} k(\boldsymbol{X}, \boldsymbol{X}) + \sigma^2 \boldsymbol{I} & \bullet & \bullet & \bullet \\ k(\boldsymbol{x}, \boldsymbol{X}) & k(\boldsymbol{x}, \boldsymbol{x}) & \bullet & \bullet \\ \nabla k(\boldsymbol{x}, \boldsymbol{X}) & \nabla k(\boldsymbol{x}, \boldsymbol{x}) & \nabla^2 k(\boldsymbol{x}, \boldsymbol{x}) & \bullet \\ \nabla^2 k(\boldsymbol{x}, \boldsymbol{X}) & \nabla^2 k(\boldsymbol{x}, \boldsymbol{x}) & \nabla^3 k(\boldsymbol{x}, \boldsymbol{x}) & \nabla^4 k(\boldsymbol{x}, \boldsymbol{x}) \end{bmatrix} \right) \quad (3)$$

where $\boldsymbol{y} \in \mathbb{R}^n$ are the $n$ function observations, $\boldsymbol{X} = [\boldsymbol{x}_1, \ldots, \boldsymbol{x}_n] \in \mathbb{R}^{d \times n}$ is the matrix of all training inputs with $\boldsymbol{x}_i \in \mathbb{R}^d$, and $\boldsymbol{x} \in \mathbb{R}^d$ is the test point.[1] Here and in the following, we use $\bullet$ to denote symmetric entries for improved readability. The joint conditional distribution is then:[1]

$$\begin{bmatrix} f \\ \nabla f \\ \nabla^2 f \end{bmatrix} \mid \boldsymbol{x}, \boldsymbol{X}, \boldsymbol{y} \sim \mathcal{N}\left( \begin{bmatrix} \mu_f(\boldsymbol{x}) \\ \boldsymbol{\mu}_{\nabla f}(\boldsymbol{x}) \\ \boldsymbol{\mu}_{\nabla^2 f}(\boldsymbol{x}) \end{bmatrix}, \begin{bmatrix} \sigma_f^2(\boldsymbol{x}) & \bullet & \bullet \\ \boldsymbol{\Sigma}_{\nabla f, f}(\boldsymbol{x}) & \boldsymbol{\Sigma}_{\nabla f}(\boldsymbol{x}) & \bullet \\ \boldsymbol{\Sigma}_{\nabla^2 f, f}(\boldsymbol{x}) & \boldsymbol{\Sigma}_{\nabla^2 f, \nabla f}(\boldsymbol{x}) & \boldsymbol{\Sigma}_{\nabla^2 f}(\boldsymbol{x}) \end{bmatrix} \right). \quad (4)$$

Following standard conditioning of multivariate normal distributions, we can directly compute the mean and covariance functions of the marginals of the posterior as

$$\textit{Marginal GP of } f : \begin{cases} \mu_f(\boldsymbol{x}) = m(\boldsymbol{x}) + k(\boldsymbol{x}, \boldsymbol{X})\boldsymbol{K}^{-1}(\boldsymbol{y} - \boldsymbol{m}(\boldsymbol{X})) & \in \mathbb{R}, \\ \sigma_f^2(\boldsymbol{x}) = k(\boldsymbol{x}, \boldsymbol{x}) - k(\boldsymbol{x}, \boldsymbol{X})\boldsymbol{K}^{-1}k(\boldsymbol{X}, \boldsymbol{x}) & \in \mathbb{R} \end{cases} \quad (5)$$

$$\textit{Marginal GP of } \nabla f : \begin{cases} \boldsymbol{\mu}_{\nabla f}(\boldsymbol{x}) = \nabla m(\boldsymbol{x}) + \nabla k(\boldsymbol{x}, \boldsymbol{X})\boldsymbol{K}^{-1}(\boldsymbol{y} - \boldsymbol{m}(\boldsymbol{X})) & \in \mathbb{R}^d, \\ \boldsymbol{\Sigma}_{\nabla f}(\boldsymbol{x}) = \nabla^2 k(\boldsymbol{x}, \boldsymbol{x}) - \nabla k(\boldsymbol{x}, \boldsymbol{X})\boldsymbol{K}^{-1}\nabla k(\boldsymbol{X}, \boldsymbol{x}) & \in \mathbb{R}^{d \times d} \end{cases} \quad (6)$$

Here, we defined the Gram matrix as $\boldsymbol{K} := k(\boldsymbol{X}, \boldsymbol{X}) + \sigma^2 \boldsymbol{I}$ with entries $[k(\boldsymbol{X}, \boldsymbol{X})]_{ij} = k(\boldsymbol{x}_i, \boldsymbol{x}_j)$ for $i, j \in \mathbb{I}_n$, and use the notation that $k(\boldsymbol{X}, \boldsymbol{x}) \in \mathbb{R}^{n \times 1}$ is the vector of kernel evaluations between each training point and the test point, with entries $[k(\boldsymbol{X}, \boldsymbol{x})]_i = k(\boldsymbol{x}_i, \boldsymbol{x})$. Similarly, we can obtain the covariance term $\boldsymbol{\Sigma}_{\nabla f, f}(\boldsymbol{x})$ as well as the mean estimate of the Hessian.[2] As noted in Müller et al. [43], we must perform the inversion of the Gram matrix $\boldsymbol{K}$ *only once*. So, while calculating the gradient distribution and the mean of the Hessian is not for free, the additional computational overhead is limited with increasing data set size.

## 3.3 Related work

**Scalable Bayesian optimization** For long, BO has been considered challenging for high-dimensional input spaces leading to the development of tailored algorithms for this setting. Such

---

[1]We write the joint distribution over $f$, $\nabla f$, and $\nabla^2 f$ in block matrix form to convey intuition, though this is an abuse of notation. Formally, all components are vectorized and stacked into a single multivariate normal vector. Specifically, we have $[\boldsymbol{y}, f, \nabla f^\top, \mathrm{vec}(\nabla^2 f)^\top]^\top \in \mathbb{R}^{n+1+d+d^2}$. Similar for the mean and covariance.

[2]From hereon, we will only consider the mean of the Hessian as storing the variance over all terms as well as all covariances is very computationally intensive: Let $\nabla^2 f(\boldsymbol{x}) \in \mathbb{R}^{d \times d}$, then $\mathrm{Cov}[\nabla^2 f(\boldsymbol{x})] \in \mathbb{R}^{d \times d \times d \times d}$.

approaches include LBO methods, which we will discuss in more detail in the following, as well as methods that aim to leverage a potential underlying structure or lower dimensional effective dimensionality [33, 62, 13, 47]. Recent results show that some of the core challenges in this high-dimensional setting are due to a numerical issues when optimizing the hyperparameters, which can be in part addressed by enforcing larger lengthscales [30, 67, 48]. These developments do not make scalable approaches obsolete. Rather, we see them as a tool to further improve the modeling also for scalable BO approaches. To address scalability, in the sense of scaling with data, alternative surrogates for BO such as neural networks [56, 35, 8] or sparse GPs [42, 41] have been discussed; addressing these scalability issues, however, is not the focus of this work.

**Local Bayesian optimization**  LBO methods aim to improve the efficiency of the optimization process by focusing on local regions of the search space. Approaches such as TuRBO [15] and SCBO [14] can be classified as *pseudo-local* methods: their trust-region approach still allows for the exploration of multiple local areas and only over time collapses to one local region. On the contrary, Müller et al. [43] introduced with GIBO a new paradigm of LBO combining gradient-based approaches with BO. Since then, the algorithm has been modified with different acquisition functions to actively learn the gradient [45, 58, 23, 16], theoretically investigated [65], and extended with crash constraints [61]. This class of algorithms operates fully locally. Our algorithm BayeSQP can also be classified as such a local method. In this sense, BayeSQP extends GIBO to second-order optimization by using a Hessian approximation from a GP. Similar ideas have been leveraged in a quasi-Newton methods [11]. However, by incorporating ideas from SQP, BayeSQP is directly applicable to both unconstrained *and* constrained optimization problems—something which is not possible with GIBO.

**Bayesian optimization and Gaussian processes in classical optimization**  There have been various papers integrating BO with first-order optimization, e.g., for line search [38, 57]. GPs have been successfully applied and leveraged in optimization—both for local optimization [25, 24] and global optimization (essentially BO) [32, 18]. All of these can be classified as a subfield of probabilistic numerics [27, 28]. Similar to our approach, Gramacy et al. [20] merged classical methods with BO by lifting the constraints into the objective using an augmented Lagrangian approach which later got extended to a slacked [49] and recently a relaxed version [4]. These approaches are based on expected improvement (EI) and, crucially, Eriksson and Poloczek [14] showed that these approaches do not scale well to high-dimensional problems. BayeSQP differs in the type of acquisition function for the line search as well as the framework as it builds on SQP. To our knowledge, we are the first to leverage a joint GP model of the function, its gradient and Hessian in a classical framework.

## 4  BayeSQP: Merging classic SQP and Bayesian optimization

This paper proposes the LBO approach BayeSQP. As described above, the main objects of this approach are GP models of the objective and possible constraints that jointly model the function value, the gradient as well as the Hessian in a single model. BayeSQP then leverages this model at each iteration to construct a quadratic uncertainty-aware subproblem for a search direction that yields improvement with high probability. In the following, we will first discuss our modeling approach. Based on this, we will construct the subproblem, followed by a discussion on line search. In the end, we touch on further practical extensions and give intuition on the optimization behavior.

### 4.1  Second-order Gaussian processes as surrogate models for BayeSQP

In BayeSQP, we aim to leverage ideas from both SQP and BO to solve constrained black-box optimization problems as in (1). For this, we will model the objective and all constraints using second-order GP models introduced in Section 3.2 here stated for the objective:

$$\begin{bmatrix} f \\ \nabla f \\ \mathrm{vec}(\nabla^2 f) \end{bmatrix} \mid \boldsymbol{x}, \boldsymbol{X}, \boldsymbol{y} \sim \mathcal{N}\left( \begin{bmatrix} \mu_f(\boldsymbol{x}) \\ \boldsymbol{\mu}_{\nabla f}(\boldsymbol{x}) \\ \mathrm{vec}(\boldsymbol{\mu}_{\nabla^2 f}(\boldsymbol{x})) \end{bmatrix}, \begin{bmatrix} \sigma_f^2(\boldsymbol{x}) & \bullet & \times \\ \boldsymbol{\Sigma}_{\nabla f, f}(\boldsymbol{x}) & \boldsymbol{\Sigma}_{\nabla f}(\boldsymbol{x}) & \times \\ \times & \times & \times \end{bmatrix} \right) \quad (7)$$

We use surrogate models of the same form for each constraint $c_i(\boldsymbol{x})$. We do not compute the covariance of the Hessian ($\times$) due the scaling issues with dimensions discussed in Section 3.2. Figure 2 demonstrates the effectiveness of such a joint GP model. We can estimate the gradient, identify local optima, and estimate curvature all from only zeroth-order information.

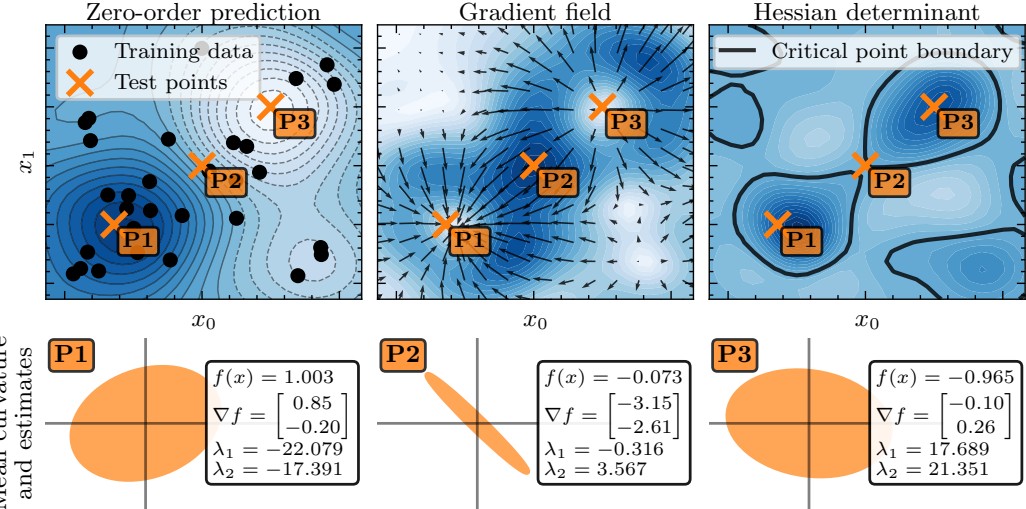

Figure 2: *The power of Gaussian processes.* Although we only have zeroth-order information about the function, the differentiability of the GP allows us to estimate both the gradient and curvature. All estimates provided are in expectation; the associated uncertainties are not shown.

Crucially, it is not required to always evaluate the full posterior distribution for each test point. In a SQP framework, we can approximate the Hessian of the Lagrangian once at our current iterate as

$$\boldsymbol{H}_t = \boldsymbol{\mu}_{\nabla^2 f}(\boldsymbol{x}_t) - \sum_{i=1}^{m} \xi_i^{(t-1)} \boldsymbol{\mu}_{\nabla^2 c_i}(\boldsymbol{x}_t), \tag{8}$$

where $\xi_i^{(t-1)}$ are the Lagrange multipliers from the solution of the last subproblem, but for the subsequent line search, we can directly work with the cheap marginal GP $f \sim \mathcal{N}(\mu_f(\boldsymbol{x}), \sigma_f^2(\boldsymbol{x}))$.

### 4.2 Deriving the subproblem for BayeSQP

Standard SQP approaches typically require exact knowledge of the objective function, constraints, and their respective gradients. In our case, we only have access to zero-order feedback and the question arises how to formulate a suitable subproblem given our choice of surrogate model.

**Expected value SQP subproblem**  A straightforward approach is to simply formulate a subproblem using expectations, leading to the following expected value subproblem:

$$\begin{aligned} \boldsymbol{p}_t \in \operatorname*{arg\,min}_{\boldsymbol{p} \in \mathbb{R}^d} & \quad \mathbb{E}\left[\frac{1}{2}\boldsymbol{p}^\top \boldsymbol{H}_t \boldsymbol{p} + \nabla f(\boldsymbol{x}_t)^\top \boldsymbol{p} + f(\boldsymbol{x}_t)\right] \\ \text{subject to} & \quad \mathbb{E}\left[c_i(\boldsymbol{x}_t) + \nabla c_i(\boldsymbol{x}_t)^\top \boldsymbol{p}\right] \geq 0, \ \forall i \in \mathbb{I}_m \,. \end{aligned} \tag{9}$$

While intuitive, this formulation fails to account for the inherent uncertainty in the estimates. As discussed by Nguyen et al. [45] and He et al. [23], taking into account the uncertainty of, e.g., the gradient, can be crucial for improving with high probability for LBO approaches.

**Uncertainty-aware SQP subproblem**  To address this limitation, we reformulate the standard QP subproblem into a robust version that explicitly accounts for uncertainty in the estimates:

$$\begin{aligned} \boldsymbol{p}_t \in \operatorname*{arg\,min}_{\boldsymbol{p} \in \mathbb{R}^d} & \quad \underbrace{\operatorname{VaR}_{1-\delta_f}\left[\frac{1}{2}\boldsymbol{p}^\top \boldsymbol{H}_t \boldsymbol{p} + \nabla f(\boldsymbol{x}_t)^\top \boldsymbol{p} + f(\boldsymbol{x}_t)\right]}_{\text{Objective value-at-risk with confidence level } 1-\delta_f} \\ \text{subject to} & \quad \underbrace{\mathbb{P}\left(c_i(\boldsymbol{x}_t) + \nabla c_i(\boldsymbol{x}_t)^\top \boldsymbol{p} \geq 0\right) \geq 1 - \delta_c}_{\text{Constraint satisfaction with confidence } 1-\delta_c}, \ \forall i \in \mathbb{I}_m \,. \end{aligned} \tag{10}$$

This formulation accounts for uncertainty through two mechanisms: employing value-at-risk (VaR) for the objective function and enforcing probabilistic feasibility for the constraints. The resulting

search direction minimizes the worst-case objective value while ensuring the constraints are satisfied with high probability.

**Tractability through joint Gaussian process**   The robust formulation in (10) remains intractable without distributional assumptions. By modeling the objective and constraints as jointly Gaussian with their gradients, we can transform (10) into a deterministic second-order cone program. Next, we derive this tractable reformulation for the constraints; the objective follows analogously.

For the constraints, we aim to ensure that $\mathbb{P}\left(c_i(\boldsymbol{x}_t) + \nabla c_i(\boldsymbol{x}_t)^\top \boldsymbol{p} \geq 0\right) \geq 1 - \delta_c$. Since we have $\boldsymbol{z}^\top \boldsymbol{v} \sim \mathcal{N}(\boldsymbol{\mu}_z^\top \boldsymbol{v}, \boldsymbol{v}^\top \boldsymbol{\Sigma}_z \boldsymbol{v})$ for a multivariate Gaussian random variable $\boldsymbol{z} \sim \mathcal{N}(\boldsymbol{\mu}_z, \boldsymbol{\Sigma}_z)$ and $\boldsymbol{v}$ is a deterministic vector, we know that $c_i(\boldsymbol{x}_t) + \nabla c_i(\boldsymbol{x}_t)^\top \boldsymbol{p}$ is also normal distributed with moments

$$\mathbb{E}\left[c_i(\boldsymbol{x}_t) + \nabla c_i(\boldsymbol{x}_t)^\top \boldsymbol{p}\right] = \mu_{c_i}(\boldsymbol{x}_t) + \boldsymbol{\mu}_{\nabla c_i}^\top(\boldsymbol{x}_t)\,\boldsymbol{p} \tag{11}$$

$$\mathrm{Var}\left[c_i(\boldsymbol{x}_t) + \nabla c_i(\boldsymbol{x}_t)^\top \boldsymbol{p}\right] = \sigma_{c_i}^2(\boldsymbol{x}_t) + \boldsymbol{p}^\top \boldsymbol{\Sigma}_{\nabla c_i}(\boldsymbol{x}_t)\,\boldsymbol{p} + 2\boldsymbol{p}^\top \boldsymbol{\Sigma}_{c_i,\nabla c_i}(\boldsymbol{x}_t) \tag{12}$$

where the last term accounts for the covariance between the function and its gradient. In the following, we drop the explicit evaluation at $\boldsymbol{x}_t$ for all moments for notational convenience, i.e., $\mu_{c_i} = \mu_{c_i}(\boldsymbol{x}_t)$.

For a Gaussian random variable to remain non-negative with probability at least $1 - \delta$, we require its mean to exceed its standard deviation multiplied by the corresponding quantile. This yields:

$$\mu_{c_i} + \boldsymbol{\mu}_{\nabla c_i}^\top \boldsymbol{p} \geq q_{1-\delta}\sqrt{\sigma_{c_i}^2 + \boldsymbol{p}^\top \boldsymbol{\Sigma}_{\nabla c_i}\,\boldsymbol{p} + 2\boldsymbol{p}^\top \boldsymbol{\Sigma}_{c_i,\nabla c_i}} \tag{13}$$

where $q_{1-\delta} = \Phi^{-1}(1 - \delta)$ denotes the $(1-\delta)$-quantile of the standard normal distribution. Rearranging the terms and introducing an auxiliary variable $t_{c_i}$ to upper-bound the square root term allows the constraint to be reformulated as a set of two inequalities:

$$-\boldsymbol{\mu}_{\nabla c_i}^\top \boldsymbol{p} + q_{1-\delta}b_{c_i} \leq \mu_{c_i} \quad \text{and} \quad \sqrt{\sigma_{c_i}^2 + \boldsymbol{p}^\top \boldsymbol{\Sigma}_{\nabla c_i}\,\boldsymbol{p} + 2\boldsymbol{p}^\top \boldsymbol{\Sigma}_{c_i,\nabla c_i}} \leq b_{c_i} \tag{14}$$

To express the square root term more compactly, we consider the full covariance matrix associated with the joint Gaussian distribution of $c_i$ and its gradient $\nabla c_i$. Specifically, we can state

$$\sigma_{c_i}^2 + \boldsymbol{p}^\top \boldsymbol{\Sigma}_{\nabla c_i}\,\boldsymbol{p} + 2\boldsymbol{p}^\top \boldsymbol{\Sigma}_{c_i,\nabla c_i} = \begin{bmatrix}1 \\ \boldsymbol{p}\end{bmatrix}^\top \begin{bmatrix} \sigma_{c_i}^2 & \bullet \\ \boldsymbol{\Sigma}_{c_i,\nabla c_i} & \boldsymbol{\Sigma}_{\nabla c_i} \end{bmatrix} \begin{bmatrix}1 \\ \boldsymbol{p}\end{bmatrix} \tag{15}$$

By Cholesky decomposition of the covariance matrix, we can express the square root term as a second-order cone constraint:

$$\sqrt{\sigma_{c_i}^2 + \boldsymbol{p}^\top \boldsymbol{\Sigma}_{\nabla c_i}\,\boldsymbol{p} + 2\boldsymbol{p}^\top \boldsymbol{\Sigma}_{c_i,\nabla c_i}} = \sqrt{\begin{bmatrix}1 \\ \boldsymbol{p}\end{bmatrix}^\top \boldsymbol{L}_{c_i}\boldsymbol{L}_{c_i}^\top \begin{bmatrix}1 \\ \boldsymbol{p}\end{bmatrix}} = \left\|\boldsymbol{L}_{c_i}^\top \begin{bmatrix}1 \\ \boldsymbol{p}\end{bmatrix}\right\|_2 \leq b_{c_i} \tag{16}$$

Using the same reasoning, we can reformulate the objective function by introducing the auxiliary variable $b_f$. In the end, we obtain the following formulation that we refer to as **B-SUB**.

---

**The uncertainty-aware subproblem of BayeSQP**                                      (**B-SUB**)

$$\boldsymbol{p}_t \in \underset{\boldsymbol{p}, b_f, \{b_{c_i}\}_{i \in \mathbb{I}_m}}{\arg\min} \quad \frac{1}{2}\boldsymbol{p}^\top \boldsymbol{H}_t \boldsymbol{p} + \boldsymbol{\mu}_{\nabla f}^\top \boldsymbol{p} + \mu_f + q_{1-\delta_f}b_f \tag{17}$$

$$\text{subject to} \quad \left\|\boldsymbol{L}_f^\top \begin{bmatrix}1 \\ \boldsymbol{p}\end{bmatrix}\right\|_2 \leq b_f, \quad \left\|\boldsymbol{L}_{c_i}^\top \begin{bmatrix}1 \\ \boldsymbol{p}\end{bmatrix}\right\|_2 \leq b_{c_i}, \quad \forall i \in \mathbb{I}_m,$$

$$-\boldsymbol{\mu}_{\nabla c_i}^\top \boldsymbol{p} + q_{1-\delta_c}b_{c_i} \leq \mu_{c_i}, \qquad \forall i \in \mathbb{I}_m.$$

where $\boldsymbol{L}_f$ and $\boldsymbol{L}_{c_i}$ are Cholesky factorizations as

$$\boldsymbol{L}_f \boldsymbol{L}_f^\top = \begin{bmatrix} \sigma_f^2 & \bullet \\ \boldsymbol{\Sigma}_{\nabla f,f} & \boldsymbol{\Sigma}_{\nabla f} \end{bmatrix}, \quad \boldsymbol{L}_{c_i}\boldsymbol{L}_{c_i}^\top = \begin{bmatrix} \sigma_{c_i}^2 & \bullet \\ \boldsymbol{\Sigma}_{\nabla c_i,c_i} & \boldsymbol{\Sigma}_{\nabla c_i} \end{bmatrix}, \quad \forall i \in \mathbb{I}_m. \tag{18}$$

*We omitted the explicit dependency on $\boldsymbol{x}_t$ for clarity but all moments are evaluated at $\boldsymbol{x}_t$.*

---

This formulation also naturally incorporates the subproblem formulation in (9).

**Corollary 1** (Recovering the expected value formulation). *The solution for the search direction of* **B-SUB** *is equivalent to solution of* (9) *for* $\delta_f = 0.5$ *and* $\delta_c = 0.5$.             (Proof in Appendix C)

**Remark 1.** *In practice, the numerical solver will have an influence on the obtained results. So while the cones no longer restrict the search direction, a cone solver might still return a different solution.*

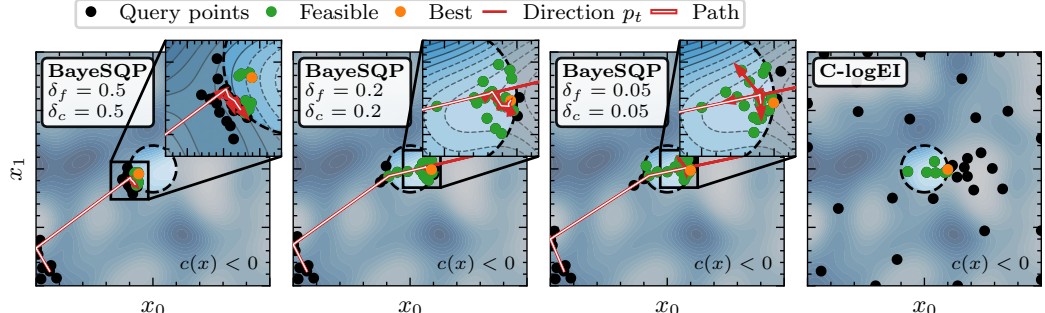

Figure 3: *Intuition on optimization behavior of* BayeSQP. *Disregarding uncertainty ($\delta_f, \delta_c = 0.5$, left) results in directions tangential to the circular constraint, while a for a very conservative configuration ($\delta_f, \delta_c = 0.05$, center right), the constraint acts as a repellent to ensure feasibility. Values in between (center left) yield a desirable convergence path to the optimum. On the right, we see the space-filling behavior of constrained logEI which is fundamentally different compared to the local* BayeSQP.

## 4.3 Line search through constrained posterior sampling

With the search direction given as the solution of the **B-SUB** subproblem, the next step is to decide on a step size $\alpha$ which which we can update the current iterate as $\boldsymbol{x}_{t+1} = \boldsymbol{x}_t + \alpha_t \boldsymbol{p}_t$. To implicitly decide on the step size, we perform constrained posterior sampling [14] on the one-dimensional line segment spanned by $\boldsymbol{p}_t$. Specifically, we aim to solve

$$\underset{\{\boldsymbol{x}_t + \alpha \boldsymbol{p}_t \mid \alpha \in [0,1]\}}{\arg \min} \quad f(\boldsymbol{x}) \quad \text{subject to} \quad c_i(\boldsymbol{x}) \geq 0, \ \forall i \in \mathbb{I}_m. \tag{19}$$

This is similar to LineBO [34] but for an objective under potentially multiple constraints. However, in contrast to LineBO, our approach does not attempt global convergence along the line. Instead, we aim to select a sufficiently promising $\alpha_t$ that yields progress given a limited evaluation budget $M$ for the line search which we set to 3 in all experiments. Similar to [14], we either choose the next point to be the index of the best feasible point, or, if none of the points are feasible, as the point with the least amount of constraint violations as

$$\boldsymbol{x}_{k+1} \leftarrow \begin{cases} \arg \min_{\boldsymbol{x}_t^{(j)} \in \mathcal{F}} f(\boldsymbol{x}_t^{(j)}), & \text{if } \mathcal{F} \neq \emptyset, \\ \arg \min_{1 \leq j \leq M} \sum_{i \in \mathbb{I}_m} \max \left(0, -c_i\left(\boldsymbol{x}_t^{(j)}\right)\right), & \text{otherwise,} \end{cases} \tag{20}$$

where $\mathcal{F} = \left\{ \boldsymbol{x}_t^{(j)} \, \middle| \, c_i\left(\boldsymbol{x}_t^{(j)}\right) \geq 0, \ \forall i \in \mathbb{I}_m \right\}$ denotes the set of feasible points among the $M$ samples.

## 4.4 Practical considerations and intuition on optimization behavior

**Local sub-sampling** Unlike GIBO-style methods [43, 45, 23], we decide against adaptive sub-sampling which would require optimizing over the uncertainty of the Hessian which is computationally very expensive. Instead, to approximate local curvature after each line search, we sample $K$ points from a $d$-dimensional ball of radius $\varepsilon$ centered at $\boldsymbol{x}_t \in \mathbb{R}^d$. For this, we first draw a Sobol sequence from the hypercube $[0, 1]^{d+1}$. Each Sobol point $(\tilde{\boldsymbol{x}}, u) \in [0, 1]^d \times [0, 1]$ is then transformed such that $\tilde{\boldsymbol{x}}$ approximates a standard normal vector to yield a unit direction $\bar{\boldsymbol{x}}$, and $u$ determines the individual radius as $r = \varepsilon \cdot u^{1/d}$. The final sample is then $\boldsymbol{x} = \boldsymbol{x}_t + r \cdot \bar{\boldsymbol{x}}$.

**Slack variable fallback strategy** The subproblem **B-SUB** may become infeasible due to constraint linearization or high uncertainty in gradient estimates. To address this, we implement a slack variable version of **B-SUB** as a fallback, which guarantees feasibility by design (cf. Appendix E). This approach aligns with established practices in classical SQP methods [46]. While the resulting search direction may not provide optimal robustness against uncertainty, the constrained posterior sampling along this direction will still seek to improve upon the current iterate.

**Intuition on optimization behavior** To gain intuition about the parameters $\delta_f$ and $\delta_c$ and their influence on the optimization process, we study BayeSQP on a small toy example. We generate

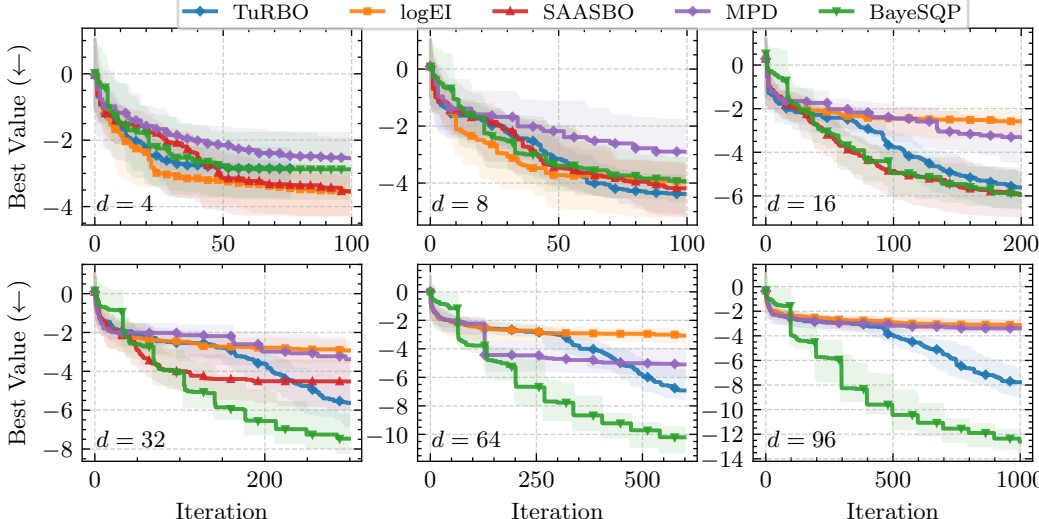

Figure 4: *Unconstrained within-model comparison.* As the dimensions grow, the benefit of local search increases, with BayeSQP significantly outperforming the other baselines. Note that for SAASBO, no runs completed within the 24-hour time cap when the dimensionality exceeded 32.

a two-dimensional within-model objective function (cf. Appendix A) with a quadratic constraint, resulting in only a small feasible region in the center. Figure 3 illustrates the optimization paths for different parameterizations. The initial step from the bottom left appears identical for all parameter settings. Subsequently, however, their behaviors differ significantly. In the expected value formulation ($\delta_f, \delta_c = 0.5$), the linearization of the quadratic constraint results in tangential directions $p_k$, leading to limited or no improvement. We observe that incorporating uncertainty into the subproblem pushes the search direction toward the feasible set. Additionally, selecting a very low value for $\delta_c$ effectively robustifies the constraints, as shown by the resulting directions $p_k$.

## 5 Empirical evaluations

We next quantitatively evaluate our proposed method BayeSQP. Our evaluation first considers unconstrained and then constrained optimization problems using BoTorch [5]. We benchmark against four baselines: logarithmic EI (logEI) [1, 32], TuRBO [15], SAASBO [13], and MPD [45]. These baselines are widely used [40, 29, 51, 66] and represent complementary approaches—logEI employs a classic global optimization strategy, TuRBO implements a pseudo-local approach, SAASBO aims to automatically identify and exploit low-dimensional structure within high-dimensional search spaces through a hierarchical sparsity prior, and MPD is a fully local BO approach. Additionally, logEI and TuRBO can be readily adapted for constrained optimization through their respective variants: C-logEI [1, 17, 19] and SCBO [14] to which we compare on the constrained optimization problems.

In all subsequent plots, we present the median alongside the 5th to 95th percentile range (90% inner quantiles) computed across 32 independent random seeds. For BayeSQP, we set the hyperparameters $\delta_f, \delta_c = 0.2$ (unless stated otherwise) and $K = d + 1$, following Wu et al. [65, Corollary 1].

**Unconstrained optimization**    We first consider unconstrained within-model problems [26] for which we adapt **B-SUB** accordingly. We generate the functions using random Fourier features following [50, 64] (cf. Appendix A for all details). Optimizing such functions has gained relevance with recent advances in latent space BO [60, 22, 40], where GP priors are enforced in the latent space [52]. Figure 4 summarizes the results. BayeSQP outperforms the other baselines from dimension 16 onward. Furthermore, we can observe the step-like behavior of BayeSQP resulting from the subsampling followed by solving **B-SUB** and the subsequent line search which yields the improvement.

---

[3]All simulations were performed on the same HPC cluster with Intel Xeon 8468 Sapphire at 2.1 GHz.

[4]Results computed across runs that successfully found feasible solutions.

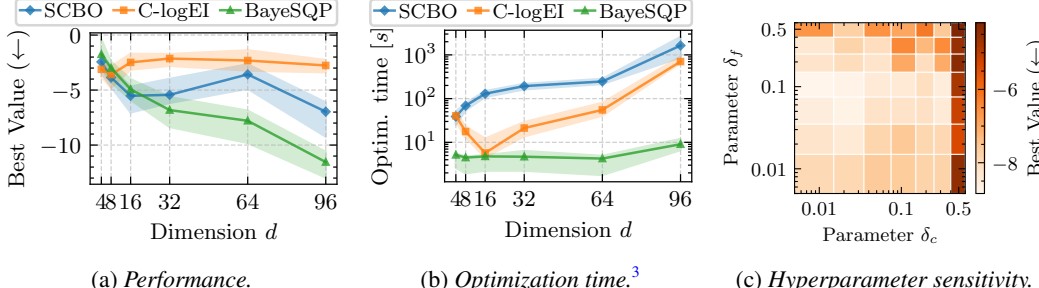

(a) *Performance.*  (b) *Optimization time.*[3]  (c) *Hyperparameter sensitivity.*

Figure 5: *Constrained within-model comparison.* `BayeSQP` demonstrates superior performance at high dimensions, fast optimization times, as well as low sensitivity to parameter choice.

Table 1: Results on popular BO benchmarks with multiple optima [14, 37]. `BayeSQP`'s local search sometimes results in worse performance but crucially it always finds feasible solutions.

| Method | Ackley5D | Hartmann | Ackley20D | Ackley5D (constr.) | Hartmann (constr.) | Ackley20D (constr.) |
|---|---|---|---|---|---|---|
| (C-)logEI | $2.47^{3.10}_{1.54}$ | $\mathbf{-3.32}^{-3.20}_{-3.32}$ | $2.83^{3.37}_{2.09}$ | $2.51^{3.04}_{1.40}$ (feas. **32 / 32**) | $-3.26^{-2.53}_{-3.32}$ (feas. **32 / 32**) | $3.41^{3.04}_{1.40}$ (feas. $15/32)^4$ |
| TuRBO / SCBO | $\mathbf{0.77}^{1.85}_{0.38}$ | $\mathbf{-3.32}^{-3.20}_{-3.32}$ | $\mathbf{2.20}^{2.58}_{1.90}$ | $\mathbf{0.51}^{1.56}_{0.18}$ (feas. **32 / 32**) | $\mathbf{-3.32}^{-2.65}_{-3.32}$ (feas. **32 / 32**) | $\mathbf{2.03}^{2.47}_{1.76}$ (feas. **32 / 32**) |
| SAASBO | $1.86^{2.29}_{1.17}$ | $\mathbf{-3.32}^{-3.20}_{-3.32}$ | $\mathbf{2.20}^{2.39}_{1.75}$ | — | — | — |
| MPD | $12.57^{14.97}_{7.74}$ | $-0.61^{-0.01}_{-2.99}$ | $13.36^{14.68}_{11.98}$ | — | — | — |
| BayeSQP | $8.95^{14.00}_{2.96}$ | $-3.30^{-1.49}_{-3.32}$ | $10.66^{11.43}_{7.57}$ | $6.25^{7.62}_{2.98}$ (feas. **32 / 32**) | $\mathbf{-3.32}^{-2.63}_{-3.32}$ (feas. **32 / 32**) | $3.90^{4.63}_{3.36}$ (feas. **32 / 32**) |

**Constrained optimization**  Similarly, we can perform within-model comparisons for the constrained case. Here, also the constraint function $c(\boldsymbol{x})$ is a sample from an GP. Again, all details are provided in Appendix A. Figure 5 summarizes the constrained within-model results. As in the unconstrained case, `BayeSQP` outperforms the baselines at high dimensions (Figure 5a), while remaining orders of magnitude faster than `SCBO` and `C-logEI` despite computing full Hessians per **B-SUB** (Figure 5b). However, as we keep increasing dimensions, computing the Hessians of size $d \times d$ will results in a computational overhead. Here, low-rank approximations might be useful for balancing the trade-off between computational efficiency and required accuracy of the subproblem—it is likely that especially in the context of BO, the accuracy of the Hessian is not of utmost importance. For a detailed runtime breakdown and discussion we refer to Appendix F. Lastly, in Figure 5c we can observe the influence of the parameters $\delta_f$ and $\delta_c$ of **B-SUB** on the performance for $d = 64$. We can observe as visualized in Figure 3, not considering uncertainty especially in the constraints ($\delta_c = 0.5$) will result in suboptimal performance for such highly non-convex constraints. Including uncertainty results in a small buffer to the boundary, allowing the algorithm to escape local optima with a small region of attraction. The figure further highlights that beyond the decisive factor of taking uncertainty into account the overall sensitivity on *how much* uncertainty should be incorporated is rather low. The optimal values of these parameters may vary depending on the specific application.

**Performance on standard benchmarks**  Lastly, we also evaluate `BayeSQP` on standard BO benchmarks. Here, we follow recent best practices and initialize lengthscales with $\sqrt{d}$ for all baselines [30, 67, 48]. The results are summarized in Table 1. We can clearly observe that `BayeSQP` is sensitive to initialization highlighted by the large 90% quantile especially for Ackley. This is to be expected as the algorithm is local and Ackley is very multi-modal. Still, importantly, `BayeSQP` is able to find feasible solutions for all seeds in all benchmarks contrary to `C-logEI`.

To demonstrate the real-world applicability of `BayeSQP`, we compare constrained optimization baselines on the 7-dimensional Speed Reducer benchmark [36], which minimizes the weight of a speed reducer subject to 11 mechanical design non-linear constraints (more details in Appendix A.4). The results are summarized in Table 2. All baselines are able to find feasible solutions for all seeds. `C-logEI` and `BayeSQP` show the best performance. In line with previous experiments,

Table 2: Performance on Speed Reducer [36].

| Method | Performance | Avg. runtime (s) |
|---|---|---|
| SCBO | $3006.89^{3013.28}_{3002.90}$ (feas. **32 / 32**) | 286.46 |
| c-logEI | $\mathbf{3002.81}^{3010.29}_{2996.67}$ (feas. **32 / 32**) | 3464.59 |
| BayeSQP | $\mathbf{3001.10}^{3009.30}_{2996.97}$ (feas. **32 / 32**) | **91.83** |

BayeSQP demonstrates a clear runtime advantage even in the presence of 11 constraints—each requiring separate Hessian evaluations—and a substantially larger **B-SUB**.

## 6 Discussion on limitations

While BayeSQP provides a novel framework combining classic optimization methods with BO, there are several limitations and addressing them will be interesting future research.

**Initialization matters** As with any local approach, the initialization of BayeSQP will directly influence its performance (cf. Table 1). This further becomes clear when looking at the flow field of BayeSQP generated from 1000 different initial conditions on the Gramacy benchmark [20] in Figure 6 (details in Appendix A.5). Depending on the initialization, the algorithm converges to a different local optimum of the constrained problem.

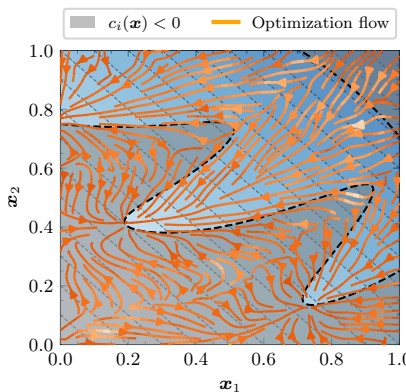

Figure 6: *Optimization behavior on Gramacy [20]*. Depending on the initialization, the BayeSQP will converge to a different local optimum. Lighter colors and thicker lines indicate larger $\|\boldsymbol{p}_k\|_2$.

Although global approaches can also exhibit sensitivity to initialization, this sensitivity is amplified in LBO approaches, particularly in constrained optimization. However, this sensitivity provides practitioners with the option to incorporate some expert knowledge into the optimization by choosing the initial guess; especially in engineering fields such as robotics, a feasible yet non-optimal solution is often known a-priori. An algorithm like BayeSQP will then become an automatic tool for fine-tuning.

**Computational considerations** We show that for up to 96 dimensions even with the additional cost of computing the Hessian, BayeSQP demonstrates as very low total runtime. Still, at very large dimensions or high number of constraints, computing as well as storing the Hessian of all constraints will become problematic. In principle, one could also incorporate Hessian uncertainty into **B-SUB**, for example following efficient schemes such as [2, 11]; whether this would lead to empirical performance improvements remains an open question. Future work could focus on evaluating the joint GP over only the *most informative* Hessian entries, adaptively selected during optimization, or on constructing the Lagrangian Hessian directly from gradient histories using a BFGS-type update scheme.

**Dependency on the kernel and model assumptions** The performance of BayeSQP strongly depends on the choice of kernel and, more generally, on the modeling assumptions underlying the GP surrogate. Since the construction of the **B-SUB** directly relies on the accuracy of both gradient and Hessian estimates, a poorly chosen kernel can lead to unreliable curvature information and ultimately to suboptimal search directions. While standard kernels such as the squared-exponential kernel perform well for smooth problems, they may struggle in settings with sharp nonlinearities or discontinuous constraints unless handled with additional care. Furthermore, kernel hyperparameters influence the scale and conditioning of the estimated Hessian, which can significantly affect the resulting search direction. Advances in GP modeling and training practices for BO (e.g., [30, 67]) are expected to directly improve the robustness and effectiveness of BayeSQP.

In Appendix B, we list possible extensions of BayeSQP which in part address the limitations mentioned above as well as further interesting directions for future work.

## 7 Conclusion

In this paper, we presented BayeSQP as a bridge between classic optimization methods and BO. BayeSQP uses GP surrogates that jointly model the function, its gradient and its Hessian, which are then used to construct subproblems in an SQP-like fashion. Our results show that BayeSQP can outperform state-of-the-art methods in high-dimensional constrained optimization problems. We believe that BayeSQP provides a promising framework for integrating well-established classical optimization principles with modern black-box optimization techniques.

## Acknowledgements

The authors thank David Stenger, Alexander von Rohr, Johanna Menn, Tamme Emunds, and Henrik Hose for various discussions on BO and optimization in general. Paul Brunzema is partially funded by the Deutsche Forschungsgemeinschaft (DFG, German Research Foundation)–RTG 2236/2 (Un-RAVeL). Simulations were performed in part with computing resources granted by RWTH Aachen University under projects rwth1579, p0022034, and p0021919.

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

# A  Design of experiments

In the following, we provide further details on the design of experiments for the experiments in Section 5 as well as discussion on the baselines and model initialization and training.

## A.1  Generating within-model objective functions

Within-model comparisons were introduced by Hennig and Schuler [26] to study the performance of BO methods on functions that fullfil all model assumptions. With recent advances in latent space BO [52], optimizing such functions has gained relevance e.g., for drug discovery. To generate the within-model objective functions shown in Figure 3 and discussed in Section 5, we approximate prior samples $f_i$ with a $M$ random Fourier features (RFFs) following Rahimi and Recht [50]. This yields a parametric function

$$f_i(\boldsymbol{x}) = \sum_{m=1}^{M} w_m \phi_m(\boldsymbol{x}) \quad \text{with} \quad \phi_m(\boldsymbol{x}) = \sqrt{\frac{2}{M}} \cos(\boldsymbol{\theta}_m^\top \boldsymbol{x} + \tau_m). \tag{21}$$

Here, $\boldsymbol{\theta}_m$ are sampled proportional to the kernel's spectral density and $\tau_m \sim \mathcal{U}(0, 2\pi)$. In all experiments, we use a squared-exponential kernel with lengthscales $\ell_i = 0.1$ and $M = 1028$ RFFs. For SAASBO, we report the out-of-model comparison results as the main mechanism of SAASBO is the way it finds suitable hyperparameters for the given task (cf. Appendix A.6 and cf. Appendix D).

## A.2  Generating constrained within-model objective functions

To generate the constrained within-model objective functions, we use the same approach as for the unconstrained case. Additionally to generating a within-model objective function, we also generate a within-model constraint function $\hat{c}(\boldsymbol{x})$. We then shift this function by one, i.e., $c(\boldsymbol{x}) = \hat{c}(\boldsymbol{x}) - 1 \geq 0$ so that on average only about 16% of the domain is feasible.[5] Note that multiple constraints are also possible, however, in the within-model setting with a shifted mean we do run the risk of generating an infeasible problem and therefore opted for only one constraint. In the other constrained benchmarks, we also consider multiple constraints.

## A.3  Constrained versions of Ackley and Hartmann

For the constrained versions of Ackley and Hartmann, we use the pre-implemented benchmarks in BoTorch [5] which are largely based on experiments in Letham et al. [37] and Eriksson and Poloczek [14]. For Ackley objectives, there are two inequality constraints and for Hartmann only one. Specifically, we use the following constraints for the Ackley and Hartmann function:

$$\text{Ackley:} \begin{cases} c_1(\boldsymbol{x}) &= -\sum_{i=1}^{d} \boldsymbol{x}_i, \\ c_2(\boldsymbol{x}) &= 5 - \|\boldsymbol{x}\|_2 \end{cases} \qquad \text{Hartmann:} \begin{cases} c_1(\boldsymbol{x}) &= 1 - \|\boldsymbol{x}\|_2^2. \end{cases} \tag{22}$$

For Ackley, we restrict the feasible region to $[-5, 10]^d$ [15] and use a time horizon of $T = 100$ for $d = 5$ and $T = 400$ for $d = 20$. The feasible region for Hartmann is $[0, 1]^6$ and we set $T = 100$.

## A.4  Speed Reducer

The Speed Reducer design problem aims to minimize the weight of a speed reducer mechanism subject to 11 mechanical constraints. The design variables are: face width ($x_1 \in [2.6, 3.6]$), module of teeth ($x_2 \in [0.7, 0.8]$), number of teeth on pinion ($x_3 \in [17, 28]$, integer which we treat as a continuous variable as implemented in BoTorch and consistent with prior work [15]), length of shaft 1 ($x_4 \in [7.3, 8.3]$), length of shaft 2 ($x_5 \in [7.8, 8.3]$), diameter of shaft 1 ($x_6 \in [2.9, 3.9]$), and diameter of shaft 2 ($x_7 \in [5.0, 5.5]$). We run the benchmark for 200 iterations across 32 random seeds with $T = 200$ and report the results in Table 2. For BayeSQP, we set $\delta_f, \delta_c = 0.5$ essentially reverting to a expected value formulation. As discussed, including uncertainty will result in the final solution being robust in the sense of not directly laying on the boundary. We find that directly at the boundary, the objective for Speed Reducer significantly improves. A very practical approach could also be to schedule $\delta_f, \delta_c$ over the number of iterations. For the full formulation of the optimization problem, we refer to the BoTorch documentation (v.13.0) as well as the original paper [36].

---

[5]We have for an output scale of one that $\mathbb{P}\{\hat{c}(x) - 1 \geq 0\} = \mathbb{P}\{Z \geq 1\} \approx 0.16$ where $Z \sim \mathcal{N}(0, 1)$.

## A.5 Constrained Gramacy function

For the constrained Gramacy benchmark, we use the formulation introduced in Gramacy et al. [20] and implemented in BoTorch. The problem is defined over the unit square $\boldsymbol{x} \in [0,1]^2$ with the objective of minimizing the sum of the two decision variables, $f(\boldsymbol{x}) = x_1 + x_2$. It imposes two nonlinear inequality constraints

$$c_1(\boldsymbol{x}) = -\left(1.5 - x_1 - 2x_2 - 0.5\sin\left(2\pi(x_1^2 - 2x_2)\right)\right), \tag{23}$$

$$c_2(\boldsymbol{x}) = -(x_1^2 + x_2^2 - 1.5). \tag{24}$$

The problem is non-convex as shown in Figure 6. For these problems, local BO approaches are especially sensitive to the initialization; depending on the initialization, different local optima may be reached. To converge to a global optimum over time, approaches such as restarting BayeSQP are promising. For a longer discussion, we refer to Appendix B.

## A.6 Baselines

**(C-)logEI** Expected improvement (EI) is a widely used acquisition function in BO. However, optimizing the EI acquisition function can be numerically unstable. To address this, Ament et al. [1] proposed a logarithmic transformation of EI resulting a numerically more stable acquisition function even resulting in an increase in performance. We use the implementation of logEI from BoTorch [5] and adapt it for constrained optimization by using the same wrapping on standard constrained EI [17, 19] resulting in C-logEI Ament et al. [1]. Neither logEI nor C-logEI require additional hyperparameters.

**TuRBO and SCBO** For the implementation of TuRBO and SCBO, we follow tutorials from BoTorch [5] which were provided by the authors of the respective methods. TuRBO (and SCBO) require various hyperparameters which specify when and by how much to shrink or expand the trust region. To set these hyperparameters, we follow the recommendations of Eriksson et al. [15] in the mentioned tutorial. With these recommendations, the initial length of the trust region is $L_{\text{init}} = 0.8$, the minimum and maximum length of the trust region are $L_{\text{min}} = 0.5^7$ and $L_{\text{max}} = 1.6$, respectively, the number of consecutive failures before the trust region is shrunk is $\tau_{\text{fail}} = \lceil \max\{4, d\} \rceil$, the number of consecutive successes before the trust region is expanded is $\tau_{\text{succ}} = 3$. The trust region is always centered around the best point found so far which in the context of constrained optimization follows (20) for SCBO. For posterior sampling, we evaluate the GP posterior within the trust region at 2000 points which are drawn from a Sobol sequence. For both methods, we use the recommended pertubation masking to cope with discrete sampling in high-dimensional spaces [54, 15], i.e., in order to not perturb all coordinates at once, we use the value in the Sobol sequence with probability $\min\{1, 20/d\}$ for a given candidate and dimension, and the value of the center of the trust region otherwise which induces an exploitation bias [48]. While TuRBO (and SCBO) depend on all the above parameters for their trust region and sampling heuristics, we found that these suggested parameters work well across various tasks as also highlighted in previous work [40, 29].

**SAASBO** Sparse axis-aligned subspace Bayesian optimization (SAASBO) [13] is designed for high-dimensional BO by placing hierarchical sparsity priors on inverse lengthscales to identify and exploit low-dimensional structure. The method uses a global shrinkage parameter $\tau \sim \mathcal{HC}(\beta)$ and dimension-specific inverse lengthscales $\rho_d \sim \mathcal{HC}(\tau)$ for all $d \in \mathbb{I}_d$, where $\mathcal{HC}$ denotes the half-Cauchy distribution. This prior encourages small values while allowing heavy tails that enable relevant dimensions to escape shrinkage toward zero. We follow the BoTorch tutorial[6] which performs inference using Hamiltonian Monte Carlo (HMC) with the NUTS sampler from Pyro [7] and uses logEI [1] as acquisition function. SAASBO's computational cost scales cubically with the number of observations due to HMC resulting a significant computational scaling as shown in Figure 8.

**MPD** Local BO via maximizing probability of decent (MPD) [45] is a follow-up method to the discussed GIBO approach. Unlike GIBO, it defines a different acquisition function for the sub-sampling step that aims to maximize the probability that the posterior mean points in a descent direction, rather than minimizing the uncertainty about the gradient. Additionally, it reuses the estimated posterior gradient GP to iteratively move the current point along the most probable descent direction

---

[6]Available under the MIT license at https://botorch.org/docs/tutorials/saasbo/ (BoTorch version v0.15.1).

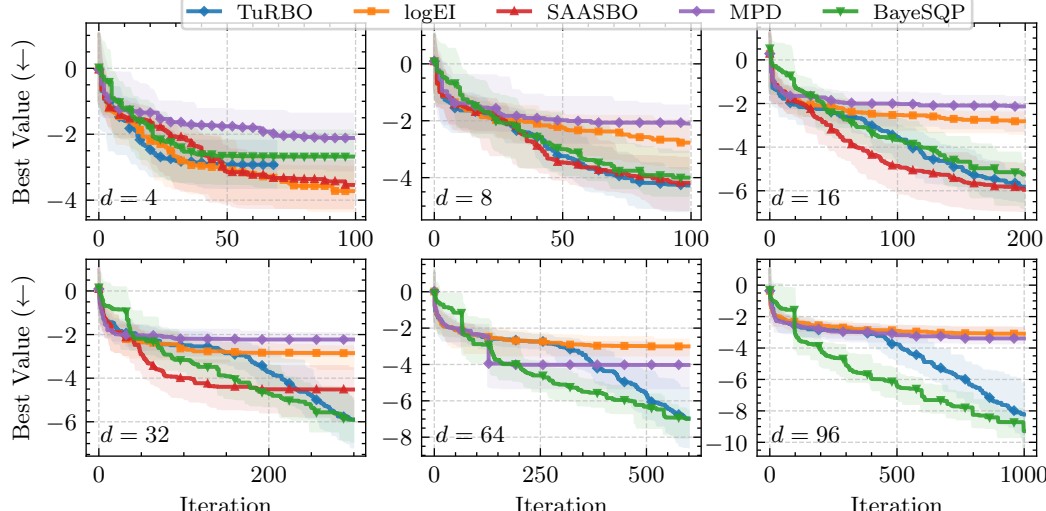

Figure 7: *Unconstrained out-of-model comparison.* As the dimensions grow, the benefit of local search increases, with BayeSQP outperforming the other baselines.

until the probability falls below a predefined threshold. We use the parameters implemented for the synthetic functions from the respective repository. [7] We should note that we did not further tune these parameters to our specific problems. We hypothesize that some of the sub-optimal performance can be attributed to the second stage of the algorithm, where the current GP estimate may be trusted for too many iterations before entering a sub-sampling step.

**BayeSQP** For BayeSQP, we set the hyperparameters $\delta_f, \delta_c = 0.2$, $K = d+1$, $M = 3$, and $\varepsilon = 0.05$ in all experiments, unless stated otherwise. To solve **B-SUB** at each iteration, we use CVXOPT [3] with standard parameters for maximum iterations and tolerance.[8] Until we have not reached a feasible point, we set $\delta_f = 0.5$ to focus on robust improvement of the constraints and switch to the specified value once we have observed a feasible point. For the subsequent line search, we use constrained posterior sampling as in SCBO but without the perturbation masking as we only operate on a one-dimensional line segment. On this line segment, we 100 sample points from a Sobol sequence as candidate points and choose the next sample location following Eriksson and Poloczek [14].

### A.7 Model initialization and training

For all algorithms, we use a squared-exponential kernel. This kernel is sufficiently smooth such that we can formulate the joint GP for BayeSQP as specified in (7). For the within-model comparisons, we freeze the lengthscales of the kernel and do not perform any hyperparameter optimization. For the experiments on classic benchmarks, we follow recent best-practice, wrap the squared-exponential kernel into a scale kernel and initialize all lengthscales with $\sqrt{d}$. We furthermore as corse bounds on the lengthscales as $\ell_i \in [0.001, 2d]$ for all baselines and set the noise to a small values i.e., , $\sigma^2 = 10^{-4}$. We subsequently optimize all hyperparameters by maximizing the marginal log-likelihood. In all experiments and for all baselines, we use standardization as a output transformation to improve numerical stability of the GP and normalize the inputs to the unit hypercube $[0, 1]^d$ [5].

## B    Extensions to different settings and ideas for future work

BayeSQP provides a flexible framework that can be extended to various settings. In this section, we briefly discuss some of these extensions as well as other interesting avenues for futuree work.

---

[7]The repository is under the MIT license at https://github.com/kayween/local-bo-mpd.
[8]CVXOPT is publicly available under a modified GNU GENERAL PUBLIC LICENSE.

**Termination and restarting** The question of *when to stop optimizing* in BO has gained increasing attention in recent years [39, 31, 63]. Addressing this question directly increases the practicality of an algorithm especially in the context of robotics where hardware experiments are often very expensive. For BayeSQP, we can draw inspiration from traditional SQP methods to develop an appropriate termination criterion such as stopping optimization once $\|\boldsymbol{p}_t\|_2 \leq \tau_{\text{tol}}$, i.e., when the search direction becomes sufficiently small, indicating likely convergence to a local optimum and little progress is to be expected in the subsequent line search. After termination, we can leverage ideas from TuRBO. Similar to how TuRBO restarts after trust region collapse, our algorithm can randomly reinitialize when optimization terminates, given that there remains sufficient computational budget.

**Batch optimization** Similar to TuRBO, implementing BayeSQP for batch BO is straightforward: We can utilize different initial conditions as distinct starting points for local optimization. All resulting data points can be combined into the same GP model or as in TuRBO separated in different data sets. In general, scaling local BO methods to batch optimization is particularly promising as these algorithms inherently remain confined to the local region surrounding their initialization point likely generating high-diversity batches.

**Localized GP model** To combat model mismatch on real-world problem, it is possible to include a sliding window on the training data of size $N_{\text{max}}$ as also proposed in [43, 45]. This effectively produces a purely local model, which can better capture the local structure. However, some care has to be taken here to as this might result in unstable learning of kernel hyperparameters.

**Active sub-sampling** In its current version, BayeSQP relies on a sub-sampling step to get good posterior estimates for the gradients and Hessians. While a space-filling sampling using a Sobol sequence already yields good results (cf. Section 5), an active approach to the sub-sampling—potentially with a stopping criterion—is interesting. One approach would be to build on ideas from Tang et al. [59]. However, an active sub-sampling likely will result in a much slower overall runtime so it depends on the specific application if this is desirable.

## C  Proof of Corollary 1

The proof of Corollary 1 follows directly from the fact that $q_{1-\delta}(0.5) = \Phi^{-1}(0.5) = 0$. With this, $b_f$ and $b_{c_i}$ are zero and no longer influence the constraints or objective. Since $b_f$ and $b_{c_i}$ are optimization variables in **B-SUB**, the cones can be trivially satisfied.

## D  Additional results

**Out-of-model comparisons** In addition to the within-model comparisons from Section 5, we also perform out-of-model comparisons. In this setting, the objective still satisfies the assumption that the model is a sample from a GP, but instead of passing the correct lengthscales to the models, each baseline learns these lengthscales. For all baselines SAASBO, we initialize the lengthscales with the true lengthscales of the objective. The results are summarized in Figure 7. We can observe the same trends as in the within-model comparisons thought gap in the final performance of TuRBO and BayeSQP is smaller. Looking at the optimization time of the out-of-model comparisons in Figure 8, still is apparent with BayeSQP being two orders of magnitudes faster than TuRBO for the 96 dimensional problems. As stated in the main text, for dimensions larger than 32, SAASBO failed to solve the problem at hand within the 24h time cap

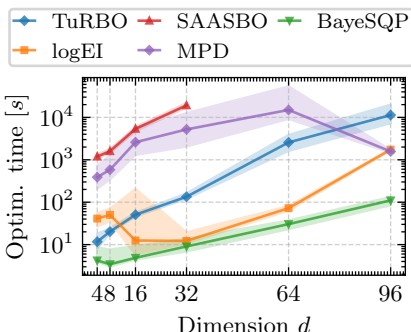

Figure 8: *Optimization time of out-of-model comparison.* BayeSQP shows significantly faster optimization compared to other baselines also in the out-of-model setting.

of the server but Figure 7 still shows the clear trend of the local approaches BayeSQP and TuRBO outperforming SAASBO in high dimensions.

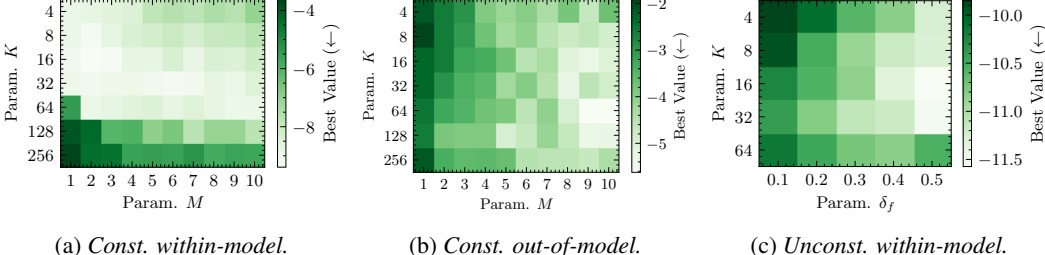

(a) *Const. within-model.*     (b) *Const. out-of-model.*     (c) *Unconst. within-model.*

Figure 9: *Ablations for* `BayeSQP`. Depending on the specific problem, different parameter combinations may yield optimal performance. All results shown in the figures correspond to an objective (and constraint) function with 64 dimensions and each field reports the median over 20 seeds.

**Ablation on the number of sub-samples** $K$ **and line search samples** $M$     Figure 5c showed that for the specified $K$ and $M$, i.e., number of sub-samples and number of line search samples, including some uncertainty in **B-SUB** will result in better performance. In Figure 9 we provide further ablations.

Figure 9a presents the results of the sensitivity analysis over the number of sub-sampling steps ($K$) and line-search steps ($M$) for both the constrained within-model and out-of-model settings. In the within-model case, increasing $K$ beyond $d$ tends to degrade performance, whereas decreasing $K$—which allows for more **B-SUB** solves under the same computational budget—can lead to improvements. For the out-of-model comparisons, the trends are less pronounced: increasing the number of line-search samples generally helps, while choosing $M$ too small can be detrimental.

Overall, the results suggest that when strong priors for the surrogate models are available (e.g., , from domain knowledge), reducing $K$ can enhance performance. In contrast, when such priors are absent, increasing $M$ may offer better results. These findings further highlight the potential benefit of introducing a suitable stopping criterion for the line search, enabling online adaptation of $M$. Finally, note that the performance of `BayeSQP` in Figure 7 could likely be improved through hyperparameter tuning—though similar improvements may be achievable for the other baselines as well.

Lastly, Figure 9c shows that, in the unconstrained within-model case, reducing $K$, as also observed in Figure 9a, can be beneficial. Moreover, even without accounting for uncertainty, the unconstrained case can achieve very good performance provided that the number of **B-SUB** solves is sufficiently large. Notably, for $K = 64$, incorporating uncertainty leads to improved results, indicating that this configuration can more effectively handle scenarios with a limited number of line-searches. The out-of-model case in Figure 10 however highlights that in the absence of good prior knowledge, reducing the number of can be costly for small $\delta_f$. A small $\delta_f$ and high uncertainty in the

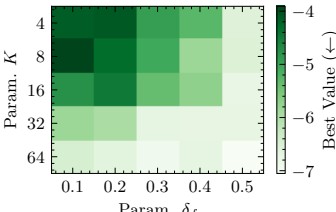

Figure 10: *Ablation for on unconstrained out-of-model functions.*

estimates in **B-SUB** will lead to small $\boldsymbol{p}_k$ and with this only limited progress towards the local optimum. A higher $K$ results in better hyperparameters and likely more confident estimates resulting in comparable process. A takeaway for practitioners in the unconstrained case is that reducing $K$ below $d$ can be advantageous. Furthermore, in the unconstrained case, employing the formulation in (9) (or equivalently setting $\delta_f = 0.5$ for **B-SUB**) can be sufficient.

## E    Numerical considerations

**Ensuring positive definiteness**     The Hessian of the Lagrangian as described in (8) may become indefinite due to numerical issues, modeling inaccuracies, or nonconvexity in the surrogate models. To maintain numerical stability and ensure that curvature information defines a valid descent direction, we enforce positive definiteness through a simple eigenvalue modification. Concretely, let $H \in \mathbb{R}^{n \times n}$ denote the Hessian candidate. We form the spectral decomposition $H = Q \Lambda Q^\top$, where $Q$ is orthogonal ($Q^\top Q = I$) and $\Lambda = \mathrm{diag}(\lambda_1, \ldots, \lambda_n)$ contains the real eigenvalues ordered arbitrarily. The eigenvalue-clipping rule replaces each eigenvalue $\lambda_i$ by $\tilde{\lambda}_i = \max(\lambda_i, \varepsilon)$ for a small threshold $\varepsilon > 0$ which we set as $\varepsilon = 10^{-5}$. The modified matrix is then reconstructed as $\tilde{H} = Q \tilde{\Lambda} Q^\top$, where

$\tilde{\Lambda} = \mathrm{diag}(\tilde{\lambda}_1, \dots, \tilde{\lambda}_n)$. By construction $\tilde{\lambda}_i \geq \varepsilon > 0$ for all $i$, hence $\tilde{H}$ is symmetric positive definite. Note that more sophisticated modifications are also possible, but we found that this simple approach already resulted in satisfactory performance.

**Jitter on the joint covariance** The joint covariance of the standard and derivative GP can be ill-conditioned. Here, we apply a standard jitter to the diagonal if necessary, which is also standard for covariances in classic BO. This then assures that the Cholesky decomposition for **B-SUB** exists.

**Ensure feasibility through slacked B-SUB formulation** After every sub-sampling step, we aim to solve the **B-SUB** optimization problem. However, given the problem and the current linearization, the sub-problem composed of the surrogate model estimates may be infeasible. We therefore opt to solve the following slack-constrained version of **B-SUB**:

$$\boldsymbol{p}_t \in \underset{\boldsymbol{p}, b_f, \{b_{c_i}\}_{i \in \mathbb{I}_m}, \{s_i\}_{i \in \mathbb{I}_m}}{\arg\min} \quad \frac{1}{2}\boldsymbol{p}^\top \boldsymbol{H}_t \boldsymbol{p} + \boldsymbol{\mu}_{\nabla f}^\top \boldsymbol{p} + \mu_f + q_{1-\delta_f} b_f + \rho \sum_{i \in \mathbb{I}_m} s_i \qquad (25)$$

$$\text{subject to} \quad \left\| \boldsymbol{L}_f^\top \begin{bmatrix} 1 \\ \boldsymbol{p} \end{bmatrix} \right\|_2 \leq b_f, \quad \left\| \boldsymbol{L}_{c_i}^\top \begin{bmatrix} 1 \\ \boldsymbol{p} \end{bmatrix} \right\|_2 \leq b_{c_i}, \quad \forall i \in \mathbb{I}_m,$$

$$-\boldsymbol{\mu}_{\nabla c_i}^\top \boldsymbol{p} + q_{1-\delta_c} b_{c_i} - s_i \leq \mu_{c_i}, \quad \forall i \in \mathbb{I}_m,$$

$$b_f \geq 0, \quad b_{c_i} \geq 0, \quad s_i \geq 0, \quad \forall i \in \mathbb{I}_m.$$

where $\boldsymbol{L}_f$ and $\boldsymbol{L}_{c_i}$ are Cholesky factorizations as

$$\boldsymbol{L}_f \boldsymbol{L}_f^\top = \begin{bmatrix} \sigma_f^2 & \bullet \\ \boldsymbol{\Sigma}_{\nabla f, f} & \boldsymbol{\Sigma}_{\nabla f} \end{bmatrix}, \quad \boldsymbol{L}_{c_i} \boldsymbol{L}_{c_i}^\top = \begin{bmatrix} \sigma_{c_i}^2 & \bullet \\ \boldsymbol{\Sigma}_{\nabla c_i, c_i} & \boldsymbol{\Sigma}_{\nabla c_i} \end{bmatrix}, \quad \forall i \in \mathbb{I}_m, \qquad (26)$$

and $\rho > 0$ is the penalty parameter for slack variables. Here, we choose $\rho = 100$. By design, this subproblem is always feasible. With the search direction from this slacked optimization problem, we proceed as described in the main part of the paper. It should further be noted that the feasibility of the problem will also depend on the $\delta_f$ and $\delta_c$. We therefore recommend that if the subproblem frequently fails to increase $\delta_f$ and $\delta_c$ and potentially use a form of scheduling for these hyperparameters.

# F   Runtime breakdown of BayeSQP

To give an idea of the computational efficiency of BayeSQP, we provide a runtime comparison against TuRBO across varying problem dimensions in Table 3. Overall, BayeSQP demonstrates a substantial reduction in total wall-clock time relative to TuRBO, particularly in higher-dimensional settings as also demonstrated in Figure 5b and Figure 8. The reported results are from the within-model setting, but training surrogate models incurs approximately the same computational cost per hyperparameter update for both methods since BayeSQP operates only with the marginal GP. Due to its sub-sampling strategy, BayeSQP requires fewer model training iterations within the same computational budget.

Table 3: Runtime comparison of BayeSQP to TuRBO across dimensions. The time of BayeSQP which is unaccounted for is due to logging overhead. A detailed per-step runtime breakdown is in Table 4.

| Dimension | TuRBO (s) | BayeSQP (s) | SOCP (s) | Hessian (s) | Subsampling (s) | TS (s) |
|---|---|---|---|---|---|---|
| 4 | 8.93±2.69 | 2.92±1.71 | 0.06±0.07 (2.2%) | 0.13±0.06 (4.6%) | 0.03±0.01 (1.0%) | 1.42±0.21 (48.5%) |
| 8 | 13.74±2.49 | 2.43±1.75 | 0.06±0.07 (2.6%) | 0.11±0.05 (4.4%) | 0.02±0.01 (1.0%) | 0.94±0.16 (38.7%) |
| 16 | 27.39±5.60 | 2.74±1.75 | 0.08±0.07 (2.8%) | 0.14±0.10 (5.3%) | 0.04±0.01 (1.4%) | 1.17±0.15 (42.7%) |
| 32 | 39.24±4.18 | 2.80±1.71 | 0.08±0.07 (2.9%) | 0.14±0.07 (5.1%) | 0.10±0.08 (3.6%) | 1.10±0.16 (39.3%) |
| 64 | 86.02±16.24 | 2.98±1.65 | 0.09±0.06 (3.1%) | 0.22±0.08 (7.2%) | 0.12±0.06 (4.0%) | 1.21±0.16 (40.5%) |
| 96 | 310.52±86.77 | 6.72±2.30 | 0.26±0.06 (3.9%) | 0.53±0.33 (7.8%) | 0.14±0.08 (2.1%) | 1.69±0.23 (25.1%) |

To provide deeper insight into the computational characteristics of each core component of BayeSQP, we analyze the per-step runtime costs in Table 4. This breakdown demonstrates how the computational burden shifts as problem dimensionality increases, with Thompson sampling rmaining the most expensive component but showing decreasing relative contribution in higher dimensions as the Hessian computation becomes more expensive. With an increased number of constraints, the contribution of evaluating Hessians will also further increase linearly in the number on constraints.

Table 4: Runtime breakdown per BayeSQP step.

| Dimension | SOCP (s/step) | Hessian (s/step) | Subsampling (s/step) | TS (s/step) |
|---|---|---|---|---|
| 4 | 0.0043 (3.9%) | 0.0090 (8.1%) | 0.0019 (1.7%) | 0.0954 (86.3%) |
| 8 | 0.0065 (5.6%) | 0.0110 (9.5%) | 0.0025 (2.2%) | 0.0959 (82.7%) |
| 16 | 0.0072 (5.3%) | 0.0137 (10.1%) | 0.0035 (2.6%) | 0.1111 (82.0%) |
| 32 | 0.0087 (5.7%) | 0.0154 (10.0%) | 0.0108 (7.0%) | 0.1187 (77.3%) |
| 64 | 0.0100 (5.6%) | 0.0236 (13.2%) | 0.0130 (7.3%) | 0.1323 (74.0%) |
| 96 | 0.0258 (10.0%) | 0.0521 (20.1%) | 0.0142 (5.5%) | 0.1671 (64.5%) |

