# OpenReview forum: "BayeSQP: Bayesian Optimization through Sequential Quadratic Programming"
_NeurIPS.cc/2025/Conference — NeurIPS 2025 spotlight_

### Official Review · Reviewer_khbc · 2025-06-29

**Clarity:** 3
**Significance:** 4
**Originality:** 4
**Rating:** 5
**Confidence:** 4

**Summary:**

This paper proposes BayeSQP, a method for constrained black-box optimization that unifies Sequential Quadratic Programming (SQP) with Bayesian Optimization (BO). BayeSQP models the objective and constraints using GPs that jointly encode the function value, gradient, and Hessian — all inferred from zeroth-order (function value only) observations. Each iteration constructs an uncertainty-aware subproblem, formulated as a second-order cone program (SOCP) using posterior means and covariances. A constrained Thompson sampling line search follows to select the next evaluation point. Empirical results show BayeSQP performs well in high-dimensional constrained settings (up to 96D), consistently returning feasible solutions when baselines fail.

**Questions:**

1. Can you provide runtime breakdowns (e.g., % time in SOCP solve vs. GP inference vs. sampling)?

2. Would first-order-only BayeSQP (without Hessians) perform similarly? An ablation would clarify the benefit of the full model.

3. Have you considered using initialization heuristics or multi-start strategies to mitigate local-only behavior?

4. Why not include at least one CEC-style constrained benchmark, even if surrogate mismatch is expected, to demonstrate robustness?

5. Could you expand on when the uncertainty-aware formulation (δ < 0.5) is most beneficial — e.g., high noise, narrow constraints, ill-conditioning?

**Ethical Concerns:**

["NO or VERY MINOR ethics concerns only"]

**Final Justification:**

My concerns have all been resolved, so I’ll keep my score as it is.

**Limitations:**

Yes, the authors have explicitly discussed several limitations in Section 6. However, one key limitation not explicitly discussed is the lack of runtime profiling. While wall-clock performance is reported, the paper does not isolate the cost of SOCP solving, GP inference, or Hessian computation — which could become dominant at scale. Adding such profiling would better characterize the method’s practical bottlenecks.

**Quality:**

3

**Strengths And Weaknesses:**

Strengths:

S1. The proposed method tightly couples second-order classical optimization and probabilistic modeling. Unlike first-order local BO methods like GIBO, BayeSQP explicitly leverages the structure of the Hessian and models uncertainty in all components — a rare and nontrivial contribution in BO.

S2. The subproblem is carefully reformulated to incorporate uncertainty using value-at-risk (VaR) and probabilistic constraint satisfaction, yielding an SOCP solvable via standard solvers. This derivation is mathematically clean and computationally viable.

S3. Experiments show that BayeSQP outperforms TuRBO and C-logEI in synthetic constrained GP benchmarks, especially in high dimensions. Unlike C-logEI, it consistently finds feasible solutions across all runs.

S4. The method includes fallback strategies (e.g., slack-variable variants for infeasibility), Sobol-based local resampling, and extensive consideration of hyperparameter robustness (e.g., δf, δc), showing good awareness of practical deployment.

Weaknesses:

W1. BayeSQP is a fully local method, relying on an initial point and never exploring globally. While this is common in high-dimensional BO, it risks getting trapped in poor local optima, especially in multimodal landscapes (e.g., Ackley5D). No hybrid exploration scheme (e.g., TuRBO-like restarts or multi-start) is employed.

W2. While Figure 5b reports wall-clock times for constrained within-model benchmarks, the runtime data has limited generality:
* It covers only a single problem class (synthetic GP-compatible functions with smooth constraints), and is not averaged across diverse benchmark types such as Ackley.
* It does not break down runtime by component (e.g., GP posterior inference, SOCP solving, line search).

W3. The use of within-model benchmarks is justified to study surrogate performance in ideal conditions, but limits generalizability. While standard BO functions like Ackley and Hartmann are included later (Table 1), the paper does not include more realistic constrained benchmarks, such as those from the CEC competitions. These benchmarks — although often non-smooth — are standard in constrained global optimization and would better demonstrate real-world feasibility and robustness, even if only as a stress test.
That said, the choice to avoid CEC is understandable: these benchmarks often violate the assumptions of GP-based modeling (smoothness, differentiability), and the authors state that second-order modeling would not be meaningful in such cases. Still, some acknowledgment of this limitation in the main text would help balance the paper’s scope.

W4. The paper would benefit from additional empirical studies isolating the contribution of second-order information. How much does using the Hessian improve performance over gradient-only or function-only variants? Is the uncertainty-aware formulation (δ < 0.5) materially better than the expectation-only one (δ = 0.5)? These comparisons are essential to justify the extra complexity of modeling second-order structure.

W5. While this is common in BO, the paper lacks any theoretical discussion of convergence, regret, or feasibility rates under noise. Given the probabilistic nature of the surrogate and the robustness of the SOCP formulation, it would be valuable to include even informal intuition about when the method is guaranteed to make progress.

---

> ### Author Rebuttal · Authors · 2025-07-31
>
> We thank the reviewer for the comments and time taken for the review. We address their questions and stated weaknesses below.
>
> > Q1 (and W2): Can you provide runtime breakdowns (e.g., % time in SOCP solve vs. GP inference vs. sampling)?
>
> We provide runtime breakdowns for BayeSQP on within-model comparisons across different dimensions below. Note that we identified a more efficient Hessian evaluation using Einsum notation, which significantly reduces runtime by avoiding nested loops. Accordingly, an updated Figure 5b reflects these improved runtimes.
> | Dimension | TuRBO (s) | BayeSQP (s) | SOCP (s) | Hessian eval (s) | Subsampling (s) | TS (s) |
> |-----------|----------|---------------------|----------|-------------|-----------------|--------|
> | 4 | 8.93±2.69 | 2.92±1.71 | 0.06±0.07 (2.2%) | 0.13±0.06 (4.6%) | 0.03±0.01 (1.0%) | 1.42±0.21 (48.5%) |
> | 8 | 13.74±2.49 | 2.43±1.75 | 0.06±0.07 (2.6%) | 0.11±0.05 (4.4%) | 0.02±0.01 (1.0%) | 0.94±0.16 (38.7%) |
> | 16 | 27.39±5.60 | 2.74±1.75 | 0.08±0.07 (2.8%) | 0.14±0.10 (5.3%) | 0.04±0.01 (1.4%) | 1.17±0.15 (42.7%) |
> | 32 | 39.24±4.18 | 2.80±1.71 | 0.08±0.07 (2.9%) | 0.14±0.07 (5.1%) | 0.10±0.08 (3.6%) | 1.10±0.16 (39.3%) |
> | 64 | 86.02±16.24 | 2.98±1.65 | 0.09±0.06 (3.1%) | 0.22±0.08 (7.2%) | 0.12±0.06 (4.0%) | 1.21±0.16 (40.5%) |
> | 96 | 310.52±86.77 | 6.72±2.30 | 0.26±0.06 (3.9%) | 0.53±0.33 (7.8%) | 0.14±0.08 (2.1%) | 1.69±0.23 (25.1%) |
>
> From the results above, we can observe that the active sampling-based line search (TS) is the most compute intensive part of BayeSQP, followed by the building the Hessian, and lastly solving the SOCP subproblem. The remaining runtime is attributed to logging overhead (note that with only 3sec of total runtime already 1.5sec of logging are quite significant) and initializing the GP models which we reinitialize in each iteration following various BoTorch tutorials. Maintaining the same model over iterations and appending the data should further decrease runtime. The time for initialization is however similar for all algorithms as it all is done through the same constructors from BoTorch. Also the time for logging is similar across methods. We will include this runtime breakdown in a similar form to Figure 5b in the paper. However, for this figure, we will report the time per BayeSQP steps. Note that above, for dimension 4, we have a longer runtime than for dimension 8 because the number of substeps is larger. The table below shows the runtime breakdown per substep. The percentages are based on the sum of the core components.
>
>
> | Dimension | SOCP (s/step) | Hessian eval (s/step) | Subsampling (s/step) | TS (s/step) |
> |-----------|---------------|------------------|---------------------|-------------|
> | 4 | 0.0043 (3.9%) | 0.0090 (8.1%) | 0.0019 (1.7%) | 0.0954 (86.3%) |
> | 8 | 0.0065 (5.6%) | 0.0110 (9.5%) | 0.0025 (2.2%) | 0.0959 (82.7%) |
> | 16 | 0.0072 (5.3%) | 0.0137 (10.1%) | 0.0035 (2.6%) | 0.1111 (82.0%) |
> | 32 | 0.0087 (5.7%) | 0.0154 (10.0%) | 0.0108 (7.0%) | 0.1187 (77.3%) |
> | 64 | 0.0100 (5.6%) | 0.0236 (13.2%) | 0.0130 (7.3%) | 0.1323 (74.0%) |
> | 96 | 0.0258 (10.0%) | 0.0521 (20.1%) | 0.0142 (5.5%) | 0.1671 (64.5%) |
>
> The per-step analysis reveals expected dimensional scaling patterns that provide insights into algorithmic behavior. While Hessian evaluation time increases with dimension (from 8.1% to 20.1%), absolute computation times remain efficient even at higher dimensions (0.05 seconds at 96 dimensions). We appreciate this suggestion, as the breakdown provides valuable insights into our algorithm's computational characteristics of our core components.
>
> Note that learning hyperparameters adds the same runtime for all baselines as the log marginal likelihood objective only operates on the marginal standard GP. Upon request of reviewer Br7S we now also have added a discussion on the complexity of the different parts of the algorithm.
>
> > Q2: Would first-order-only BayeSQP (without Hessians) perform similarly? An ablation would clarify the benefit of the full model.
>
> This is an interesting question and follows along the lines of the request by reviewer DWmx in W2. In the revised version, we have now added an additional first-order BO method, namely MPD [1], which has the substeps of learning the current gradient and then performing gradient descent using the predicted gradients. To an extent, BayeSQP is a second-order extension of this method. We can observe that after 8 dimensions, BayeSQP also outperforms this baseline leveraging the second-order approximation in a SQP style.
>
> > Q3: Have you considered using initialization heuristics or multi-start strategies to mitigate local-only behavior?
>
> We have not explored initialization heuristics or multi-starts but see it as an exciting future direction of research. In general, we believe that BayeSQP can be also be applied in a batched setting using strategies similar to TuRBO [2], which uses for each entry in a batch a separate trust region.
>
> > Q4 (and W3): Why not include at least one CEC-style constrained benchmark, even if surrogate mismatch is expected, to demonstrate robustness?
>
> Thank you for the comment. By including Ackley, we aimed to demonstrate our method's limitations on such “adversarial” objective functions for purely local approaches. For this revision, we prioritized implementing two additional baselines (SAASBO as requested by reviewer Br7S, and MPD discussed above) as well as performing ablations to further provide insights into the behavior of BayeSQP. As you note, the CEC-style functions are often stated in a way in which these local approaches tend to yield not meaningful results. We will follow your advice and further expand the discussion of BayeSQP's limitations in such settings within the main paper.
>
> > Q5: Could you expand on when the uncertainty-aware formulation (δ < 0.5) is most beneficial — e.g., high noise, narrow constraints, ill-conditioning?
>
> We see the benefits of our uncertainty-aware formulation in two different mechanisms. First, incorporating uncertainty typically improves optimization behavior by providing more reliable search directions when gradient estimates are uncertain. Second, the choice of $\delta$ will have a direct influence on the robustness of the final solution. The magnitude of $\delta$ will likely depend on the application at hand and will mostly influence the final optimal solution. If, e.g., one aims to find a robust constraint satisfying solution one can directly encode this through the choice of very small $\delta$. By design, it will yield a solution that has some safety margin. This touches also on the high-noise setting, where we naturally want some margin to the constraint. With Figure 3, we aimed to provide some qualitative insights on the influence of delta.
>
> ---
> ### References
>
> [1] Nguyen, Quan, et al. "Local Bayesian optimization via maximizing probability of descent." Advances in neural information processing systems 35 (2022): 13190-13202.
>
> [2] Eriksson, David, et al. "Scalable global optimization via local Bayesian optimization." Advances in neural information processing systems 32 (2019).

---

### Official Review · Reviewer_DWmx · 2025-06-30

**Clarity:** 2
**Significance:** 2
**Originality:** 3
**Rating:** 5
**Confidence:** 4

**Summary:**

This paper introduces BayeSQP, a novel approach that blends ideas from Sequential Quadratic Programming (SQP) and Bayesian Optimization (BO) to tackle constrained black-box optimization problems. The core idea is to build joint Gaussian Process (GP) surrogates that model not just the function values, but also their gradients and Hessians, estimated purely from zero-order observations. At each iteration, BayeSQP constructs a local subproblem that incorporates model uncertainty and solves it as a second-order cone program (SOCP) to determine a promising search direction. A subsequent constrained Thompson sampling line search is used to select the next evaluation point. The approach is tested on synthetic problems, showing a competive performance.

**Questions:**

1. *Line search budget and efficiency:* The method uses constrained Thompson sampling with a fixed number of samples (M=3) for line search. Could the authors elaborate on how sensitive BayeSQP is to this choice? Would a larger budget significantly improve performance?

2. *Alternative uses of uncertainty:* The paper leverages uncertainty using a Value-at-Risk (VaR) formulation, which is both intuitive and computationally tractable. However, one could imagine alternative approaches, such as using Conditional Value-at-Risk (CVaR) or distributionally robust optimization formulations. Have the authors considered or tested such alternatives?

3. *Guidance on uncertainty parameters:* Although the authors include an illustrative analysis of the role of $\delta_f$ and $\delta_c$, a more systematic sensitivity study would be helpful. Providing practical recommendations for setting these parameters could make the method more accessible to non-expert users.

4. *Clarification on problem formulation* In Equation (1), the use of the expectation $\mathbb{E}[f(x)]$ may be misleading, especially since the rest of the paper assumes a deterministic function with noisy observations. Could the authors clarify the meaning of this expectation, or consider reformulating the objective if the method is not intended to handle true stochastic optimization problems?

**Ethical Concerns:**

["NO or VERY MINOR ethics concerns only"]

**Final Justification:**

I believe this paper makes a significant contribution. Moreover, the authors were able to address most of my concerns during the discussion period, so I have decided to raise my score from 4 to 5.

**Limitations:**

Yes.

**Paper Formatting Concerns:**

None.

**Quality:**

3

**Strengths And Weaknesses:**

**Strenghts:**

1. *Technical depth and novelty:* The paper presents a well-thought-out integration of classical second-order optimization with Bayesian optimization. To my knowledge, this is the first work to take this approach in a probabilistic SQP framework.

2. *Strong empirical performance:* BayeSQP outperforms strong baselines like TuRBO and logEI on several high-dimensional benchmarks, particularly when constraints are involved.

**Weaknesses:**

1. *Lack of real-world benchmarks:* The experiments are well-executed, but they’re limited to synthetic functions. Including one or two real-world applications would help demonstrate the practical impact and usability of the method.

2. *Baseline selection should be broader:* The comparison focuses on established baselines like TuRBO and logEI, but doesn’t include some newer methods for local BO, such as the algorithm proposed by Nguyen et al. (2022). A broader comparison would help position BayeSQP more clearly in the current landscape.

---

> ### Author Rebuttal · Authors · 2025-07-31
>
> We thank the reviewer for the comments and time taken for the review. We address their questions and stated weaknesses below.
>
> > Q1: The method uses constrained Thompson sampling with a fixed number of samples (M=3) for line search. Could the authors elaborate on how sensitive BayeSQP is to this choice? Would a larger budget significantly improve performance?
>
> Thank you for the interesting question. We performed a sweep over different values with $M \in \\{ 1, …, 10\\}$ to test the influence on the final performance on the 64D constrained within model objective functions. We find that for a within-model setting the median performance does not significantly change but by increasing $M$ the 90% upper quantile of the performance can be reduced. In general, we hypothesize that there is a deep connection of $M$ and the number of subsampling steps $K$. We therefore conducted a second sweep over combinations (similar to Figure 5c) with $K \in \\{4, 8, 16, 32, 64, 128, 256\\}$ and found that one can reduce the number of subsampling steps when slightly increasing $M$ for a “pure” within-model (WM) setting, i.e., with perfect model knowledge. We performed the same sweep also with online hyperparameter optimization (OHO). Here, we find that increasing $M$ can improve performance but decreasing $K$ comes at a higher cost as a higher number of subsamples facilitates better learning of hyperparameters. Increasing $K$ beyond D also does not improve performance as it reduces the number of line searches given a fixed budget. That said, using D+1 subsampling points is a compelling heuristic as it offers a clean theoretical justification, as discussed in the paper, however, if more inductive biases are available one can probably reduce this number. We will include these additional sensitivity ablations in the main body of the paper.
>
> > Q2: However, one could imagine alternative approaches, such as using Conditional Value-at-Risk (CVaR) or distributionally robust optimization formulations. Have the authors considered or tested such alternatives?
> We have not yet experimented with CVaR or distributionally robust formulations, but we agree that this is a promising direction. We see BayeSQP as a flexible framework that could benefit from various extensions and refinements, including improvements to the subproblem formulation. In particular, incorporating risk-sensitive objectives like CVaR or potentially leveraging approximate Hessian uncertainties for more informed subsampling are exciting directions which seem promising for future research. We hope that this initial version of BayeSQP can serve as a foundation for further research in these directions.
>
> > Q3: Although the authors include an illustrative analysis of the role of  and , a more systematic sensitivity study would be helpful. Providing practical recommendations for setting these parameters could make the method more accessible to non-expert users.
>
> Thank you for that comment. With the illustrative example, we wanted to provide intuition on the behavior of BayeSQP for different confidence parameters and then highlight with Figure 5 c) that the exact choice of the delta does not seem to be critical as long as some delta_c < 0.5 is specified. Still, we think that we can further improve in explicit recommendations for practitioners as well as researchers. We believe that the above mentioned results will help with this and we will make sure to state suggestions more explicitly in a final version of the manuscript.
>
> > Q4: Could the authors clarify the meaning of this expectation, or consider reformulating the objective if the method is not intended to handle true stochastic optimization problems?
>
> This was a mistake from our side and thank you for spotting this! We have fixed this in the manuscript. To clarify:  we assume a fixed f from which we can only obtain noisy zero-order observations.
>
> > W1: The experiments are well-executed, but they’re limited to synthetic functions. Including one or two real-world applications would help demonstrate the practical impact and usability of the method.
>
> We agree that such an application would further strengthen the contributions of our paper. Unfortunately, we were not able to set up a new experiment in the limited time of the rebuttal and rather focused on adding baselines (MPD as requested in W2 and SAASBO requested by reviewer Br7S to which we also refer to for a longer discussion on this baseline) as well as conducting experiments that further provide insights into the behavior of BayeSQP as discussed in Q1. We also added additional out-of-model comparisons (similar to the mentioned experiments in Q1) to highlight BayeSQPs performance also in the OHO setting. In these experiments, we observe a slight drop in performance, which is to be expected, but the rankings among the algorithms remains consistent with BayeSQP showing the best performance in the high-dimensional setting.
>
> > W2: The comparison focuses on established baselines like TuRBO and logEI, but doesn’t include some newer methods for local BO, such as the algorithm proposed by Nguyen et al. (2022). A broader comparison would help position BayeSQP more clearly in the current landscape.
>
> Thank you for the comment and suggestion. We agree that including more recent local BO methods would better position BayeSQP within the current landscape and can improve the empirical section of our paper. Adding the method from the paper above is especially interesting as, in a sense, BayeSQP is an extension of this class of first-order BO methods to a second order method. We therefore added MPD (method from the mentioned paper). We find that BayeSQP outperforms this baseline in all experiments with dimension greater than 8 and we furthermore observe that in the high-dimensional experiments, also TuRBO outperformed this baseline. We should also note that extending MPD (and similar methods like GIBO) to constrained optimization settings is not straightforward.

---

> > ### Comment · Reviewer_DWmx · 2025-08-04
> >
> > I would like to thank the authors for their detailed response. Most of my concerns have been adequately addressed, and I especially appreciate the inclusion of new baselines. That said, I would like to reiterate that incorporating more realistic test problems would significantly strengthen the work. Are the authors considering adding such examples in their revision?

---

> > > ### Author Response · Authors · 2025-08-08
> > >
> > > We are glad that we were able to address most of your concerns.
> > >
> > > Regarding additional problems: We are in the process of also including the 2D Gramacy function [1] as an example to demonstrate the local nature of our algorithm. Here, we are visualizing the flow field, i.e., given an initial condition, how and to which local optimum does the algorithm converge to. We hope that this, in addition to Figure 3, will further increase intuition on BayeSQP. We are also working on including the Speed Reducer benchmark [2] (also used in [3]) into the main body of the manuscript. This benchmark is a 7D real-world problem with 11 constraints where the objective is to minimize weight the weight of the speed reducer while respecting various mechanical constraints. We can report the following results over 32 seeds showing very similar final performance across all methods:
> > >
> > > - SCBO: median 3007.17 (q5-q95: 3001.28-3012.40), 511.34s average runtime
> > > - c-logEI: median 3004.37 (q5-q95: 2999.16-3012.35), 3385.94s average runtime
> > > - BayeSQP: median 3006.48 (q5-q95: 2996.85-3016.55), 102.88s average runtime
> > >
> > > BayeSQP shows the lowest runtime—which is consistent with our prior results. We believe including this results is useful to also demonstrate that our method can also handle 11 black-box constraints.
> > >
> > > ---
> > > ### References
> > >
> > > [1] Gramacy, Robert B., et al. "Modeling an augmented Lagrangian for blackbox constrained optimization." Technometrics 58.1 (2016): 1-11.
> > >
> > > [2] Lemonge, Afonso CC, et al. "Constrained optimization problems in mechanical engineering design using a real-coded steady-state genetic algorithm." Mecánica Computacional 29.95 (2010): 9287-9303.
> > >
> > > [3] Eriksson, David, and Matthias Poloczek. "Scalable constrained Bayesian optimization." International conference on artificial intelligence and statistics. PMLR, 2021.

---

> ### Comment · Area_Chair_ZPkX · 2025-08-04
>
> Dear Reviewer DWmx, given the authors' response, please raise any remaining questions and/or concerns in a timely fashion so the authors have a chance to reply. Thanks!

---

### Official Review · Reviewer_r1jx · 2025-07-02

**Clarity:** 3
**Significance:** 3
**Originality:** 3
**Rating:** 4
**Confidence:** 3

**Summary:**

They propose BayeSQP, a method that uses classical Sequential Quadratic Programming (SQP) ideas in Bayesian Optimization for high-dimensional problems, with or without constraints. To apply SQP in BO, they require a second-order Gaussian Process, i.e., a surrogate that returns the mean, variance, gradient, and Hessian, but uses only zero-order evaluations. This GP is used to approximate the gradient and Hessian at each iteration, which allows them to construct a robust quadratic subproblem. In this subproblem they apply Value-at-Risk to the objective and chance-constraints to the restrictions, and reformulate it as a second-order cone program (SOCP). They also implement a fallback strategy with a slack variable that guarantees initial feasibility of the subproblem. The SQP subproblem they solve indicates the direction to move in iteration p_t. Then, they determine how far to move in that direction by performing a constrained line search along the segment using Thompson Sampling with a budget of M=3 evaluations, and selecting the point with the highest feasibility. Finally, in several synthetic experiments, they demonstrate the performance of their method.

**Questions:**

Questions:

 - Could BayeSQP's strong performance simply come from its initial feasibility guarantee rather than the core algorithm itself?

 - Why does logEI converge quickly and perform poorly in high-dimensional problems?

 - How many initial points were used for logEI in these experiments?

Other comments:

 - On page 3, there is a typo: "Approaches such as SCBO [11] and SCBO [10]” (one of those should be a different method reference).

 - On page 5, Eq. 9: "E [ ∇c_i (x_t )^⊤ p ≥ −c_i (x_t ) ]" is incorrect because the expectation should not include the inequality.

 - On page 5, Eq. 11: "µ^⊤_{∇c_i} (x_t)" should be: "µ_{∇c_i} (x_t)^⊤".

 - On page 6: “on a step size α **which** which we can update”.

 - At the end of page 7: there is a “p_k” but everywhere else the iteration index is “p_t”.

 - Table 1: bold highlight the best value.

**Ethical Concerns:**

["NO or VERY MINOR ethics concerns only"]

**Final Justification:**

I think that the authors have adequately addressed my comments (initial feasibility mechanism, logEI's high-dimensional performance, number of repetitions) and the important concerns the other reviewers pointed out (inclusion of SAASBO and MPD baselines, new ablation studies, new details about complexity and scalability, and a new real-world experiment). If the authors implement all of the changes mentioned, I consider it appropriate to raise the score I originally gave them.

**Limitations:**

I have described the limitations I found in the Weaknesses section.

**Paper Formatting Concerns:**

I think there are no formatting problems.

**Quality:**

3

**Strengths And Weaknesses:**

Strenghts:

 - Their method scales effectively to high-dimensional problems, with experiments demonstrated up to around 100 dimensions

 - Their method can handle both unconstrained and constrained problems

 - They formulate an uncertainty aware SQP subproblem and integrate it into a BO algorithm.

 - They implement a fallback mechanism that guarantees feasible evaluations in the first iteration.

Weaknesses:

 - The performance of BayeSQP may come from guaranteeing feasible points at the start rather than from the rest of the pieces of its algorithm. It would be useful to see an ablation study without the fallback to isolate its effect.

 - I do not understand why logEI converges so quickly and performs poorly as dimensionality increases (its performance in d=16 and d=32 is almost the same). In the within-model experiments, fixing the lengthscale of all GPs at 0.1 might not be appropriate for logEI (even if it is the best for RFF). That small scale could limit its exploration. It would be better to follow logEI's original BoTorch configuration [4] (which is the one used in logEI [1]). Moreover, it would be better to use a Matern-5/2 ARD kernel, the one used in logEI [1].

 - The number of initial points may be affecting logEI. How many were used?

- I think the number of repetitions, 20, is insufficient. They should use at least 30, and preferably 50 or 100, as is common in the literature.


[1] Sebastian Ament, Samuel Daulton, David Eriksson, Maximilian Balandat, and Eytan Bakshy. Unexpected improvements to expected improvement for Bayesian optimization. In Advances in Neural Information Processing Systems (NeurIPS), 2023.

[4] Maximilian Balandat, Brian Karrer, Daniel R. Jiang, Samuel Daulton, Benjamin Letham, Andrew Gordon Wilson, and Eytan Bakshy. BoTorch: a framework for efficient Monte-Carlo Bayesian optimization. In Advances in Neural Information Processing Systems (NeurIPS), 2020.

---

> ### Author Rebuttal · Authors · 2025-07-31
>
> We thank the reviewer for the comments and time taken for the review. We address their questions and stated weaknesses below.
>
> > Q1 (and W1): Could BayeSQP's strong performance simply come from its initial feasibility guarantee rather than the core algorithm itself?
>
> Thank you for the question but we think there may be a misunderstanding. We think our explanation of feasibility may not have been sufficiently clear, as there are actually two distinct types of feasibility at play in BayeSQP that we should distinguish better. First, there is the feasibility of a sample point x_t with respect to the original problem constraints. Second, there is the feasibility of the SOCP subproblem that BayeSQP solves at each iteration. Upon reviewing our manuscript, we recognize that our terminology was not consistently precise in distinguishing these concepts, which we will address in a final version. To directly address the question: BayeSQP implements a slacked version of the SOCP subproblem to ensure computational feasibility of the subproblem itself, not to guarantee that the resulting sample points satisfy the original problem constraints. Therefore, BayeSQP's performance cannot be attributed to an initial feasibility guarantee for the sample points. In fact, we often have multiple iterations at the start, e.g., on the constrained Hartmann, where the sample points remain infeasible similar to the other algorithms.
>
> > Q2 (and W2): Why does logEI converge quickly and perform poorly in high-dimensional problems?
>
> LogEI's poor performance in high-dimensional settings stems primarily from its global search approach. Essentially, as nicely stated in [1], when the true objective function has small lengthscales relative to domain, global search becomes increasingly ineffective as dimensionality grows. Given a small length scale for the unit hypercube and a large number of dimensions, a global search is bound to fail to a certain extent. Therefore, as problem complexity increases, logEI's behavior increasingly resembles random search which becomes particularly evident in the "pure" within-model setting as also highlighted in prior work (e.g., [2]).
>
> To investigate whether enforcing larger lengthscales could mitigate this issue, we conducted an experiment where we initialized the lengthscales as sqrt(d) [3] on the 64D within model objective. We can observe that this does improve performance slightly however, both TuRBO and BayeSQP are able to also outperform this baseline with BayeSQP showing the strongest performance.
>
> > Q3: How many initial points were used for logEI in these experiments?
>
> For all experiments and all baselines, we use D initial points. Such a scaling of the number of initial points with the number of dimensions is quite common.
>
> > W1: The performance of BayeSQP may come from guaranteeing feasible points at the start rather than from the rest of the pieces of its algorithm. It would be useful to see an ablation study without the fallback to isolate its effect.
>
> As discussed above, we believe there might have been misunderstanding about which feasibility was meant in the submitted version when discussing the slacked version of the SOCP. We adjust the text to make the distinction more precise. If we however misunderstood the point you were making, please let us know so that we can further clarify.
>
> > W2: I think the number of repetitions, 20, is insufficient. They should use at least 30, and preferably 50 or 100, as is common in the literature.
>
> We appreciate this feedback and have increased the number of repetitions from 20 to 32, following the logEI paper [3]. We are happy to increase this number even further for the final version but opted for 32 due to the limited time frame for the rebuttal. Importantly, our results remain consistent with the results reported in the submitted version of the paper.
>
> Also thank you for spotting these typos and the recommendation for Table 1! We have fixed the typos and included bold highlighting in the manuscript!
>
> ---
> ### References
>
> [1] Wüthrich, Manuel, Bernhard Schölkopf, and Andreas Krause. "Regret bounds for Gaussian-process optimization in large domains." Advances in Neural Information Processing Systems 34 (2021): 7385-7396.
>
> [2] Müller, Sarah, Alexander von Rohr, and Sebastian Trimpe. "Local policy search with Bayesian optimization." Advances in Neural Information Processing Systems 34 (2021): 20708-20720.
>
> [3] Ament, Sebastian, et al. "Unexpected improvements to expected improvement for bayesian optimization." Advances in Neural Information Processing Systems 36 (2023): 20577-20612.

---

> > ### Author Response · Authors · 2025-08-04
> >
> > Thank you, and we are glad we were able to address your concerns.
> > We noticed that you posted your comment in reviewer Br7S's thread rather than your own review thread. We wanted to bring this to your attention, as it appears that acknowledgments need to be posted in the appropriate reviewer-specific threads for proper documentation looking at the response by reviewerr khbc. We unsure how critical this is and are therefore posting this response here to ensure the area chair is aware of your positive feedback regarding our rebuttal.
> >
> > Again, thank you for your efforts.

---

> ### Comment · Reviewer_r1jx · 2025-08-04
>
> The authors have satisfactorily addressed my concerns and those of the other reviewers I wanted to understand.

---

### Official Review · Reviewer_Br7S · 2025-07-03

**Clarity:** 3
**Significance:** 3
**Originality:** 3
**Rating:** 4
**Confidence:** 4

**Summary:**

The authors present BayesSQP, which combines sequential quadratic programming with Bayesian optimization to solve constrained optimization problems. BayesSQP capitalizes on the relationship between a Gaussian Process (GP) and its derivatives to estimate the gradient and curvature of functions using zeroth-order information (function evaluations). In doing so, the QP subproblem that needs to be solved is uncertainty-aware because it uses function approximations (GPs) that naturally express uncertainty. The method is compared with TuRBO and logEI in a simulation study and using variants of the Ackley and Hartmann test functions

**Questions:**

1. Is TuRBO incorrectly labeled as SCBO on line 96?

2. There is substantial work on scalable BO which is not discussed and never compared with. For example [1] and [3]. While the focus is on constrained optimization, there is a section with unconstrained problems, so more baselines need to be included.

3. I didn't see the assumptions on the objective and constraints in the main text. Is this included somewhere and if not can the authors please include this information?

4.It would be nice to see more baselines included to know how much benefit one gets from incorporating this over existing, potentially simpler methods. For the unconstrained problems, why not compare with methods assuming an additive structure to the function [1], SAASBO [2], or random projections [3]?

5. What is the computational complexity of the algorithm? This is not explicitly stated in the text as far as I can see.

If my questions 2-5 were addressed sufficiently, e.g. extra baselines included and complexity concerns thoroughly addressed, then I would be willing to increase my score. If I have missed anything or misunderstood any components please let me know and I will reevaluate.

[1] Kandasamy, Kirthevasan, Jeff Schneider, and Barnabás Póczos. "High dimensional Bayesian optimisation and bandits via additive models." International conference on machine learning. PMLR, 2015.

[2] Eriksson, David, and Martin Jankowiak. "High-dimensional Bayesian optimization with sparse axis-aligned subspaces." Uncertainty in Artificial Intelligence. PMLR, 2021.

[3] Wang, Ziyu, et al. "Bayesian optimization in a billion dimensions via random embeddings." Journal of Artificial Intelligence Research 55 (2016): 361-387.

**Ethical Concerns:**

["NO or VERY MINOR ethics concerns only"]

**Final Justification:**

The authors directly addressed my concerns 1-5 so I have raised my score to border line accept. I feel the paper could still use more comparisons—that's why my final score is not 5.

**Limitations:**

I think the limitations of the method in terms of assumptions and scalability concerns should be fleshed out and I have included details on this in my questions.

**Quality:**

3

**Strengths And Weaknesses:**

Strengths:
- Polished presentation. The figures and organization are good. Figures 1 and 2, in particular, are well-structured and convey their information efficiently.
- The explanation of the method follows a clear structure, and the core idea is presented in a good format.
- The simulation study (Figure 4) does an excellent job of sweeping over the problem dimensions, and the results are impressive in this setting.

Weaknesses:
- The exploration of high-dimensional Bayesian Optimization solutions is lacking and should be fleshed out. See question 2 for details.
- Assumptions need to be stated more clearly. See question 3 for details.
- Scalability concerns. See question 5 for details.

---

> ### Author Rebuttal · Authors · 2025-07-31
>
> We thank the reviewer for the comments and time taken for the review. We address their questions and stated weaknesses below.
>
> > Q1: Is TuRBO incorrectly labeled as SCBO on line 96?
>
> Yes, thank you very much for catching this! We have fixed this typo in the document.
>
> > Q2 (and Weakness 1): There is substantial work on scalable BO which is not discussed and never compared with. For example [1] and [3]. While the focus is on constrained optimization, there is a section with unconstrained problems, so more baselines need to be included.
>
> Thank you for pointing out these additional references (including [2]). We agree that there is a large body of work on making BO more scalable. In our submitted version, we focused primarily on local methods—a subset of scalable BO approaches—since BayeSQP extends prior work that leverages first-order optimization techniques. However, your comment highlights that we should expand our related work discussion. We therefore include [1-3] and subsequent approaches in the related work section, adding a paragraph on scalable BO methods before discussing the local approaches. We also provide explicit experimental comparisons by incorporating an additional baseline from this set of methods to our experiments.
>
> Here, we decided to include SAASBO in our unconstrained experiments as a popular baseline in the sparse subspace setting. We find that it outperforms BayeSQP up to 8D, performs similarly on 16D, and afterwards BayeSQP also outperforms this baseline. Thank you for suggesting this baseline in Q4.
>
> > Q3 (and Weakness 2): I didn't see the assumptions on the objective and constraints in the main text. Is this included somewhere and if not can the authors please include this information?
>
> No, you are correct that we did not explicitly formulate assumptions on the objective and constraints in our submitted version. While BayeSQP does not impose explicit structural assumptions (such as sparse subspaces) on the objective and constraints, our modeling approach does embed implicit assumptions about these functions. The sub-problem is based on the assumption that the modeling approach of the objective with a GP is reasonable. Since we rely also on derivatives of the GP, severe model misspecification could potentially harm performance more than other approaches which we in part see on the Ackley experiments. We will make sure to make this more explicit in the final version. Specifically, we will add the following to Section 2: “Here and in the following, we will assume that $f(x)$ and $c(x)$ are samples from Gaussian processes and we will base our algorithm as well as modelling approach on this assumption”. We will further add a small discussion to the existing one on limitations stating that a severe model mismatch could harm the performance of BayeSQP.
>
> > Q4: It would be nice to see more baselines included to know how much benefit one gets from incorporating this over existing, potentially simpler methods. For the unconstrained problems, why not compare with methods assuming an additive structure to the function [1], SAASBO [2], or random projections [3]?
>
> As discussed above, we agree that a discussion on these methods was missing and we now have added it to the manuscript. We furthermore agree that a comparison to one of the methods mentioned above would benefit the current presentation. Therefore, we included SAASBO as a representative recent method from this category. SAASBO matched or exceeded BayeSQP's performance up to 16 dimensions, but BayeSQP significantly outperformed SAASBO at 32 dimensions. For dimensions 64 and 96, SAASBO could not complete runs within our 24-hour computational budget. This computational limitation has also been noted in prior evaluations of SAASBO [4].
>
> > Q5 (and Weakness 3): What is the computational complexity of the algorithm? This is not explicitly stated in the text as far as I can see.
>
> The computational complexity with respect to the number of data points follows standard BO at $O(N^3)$ for Gram matrix inversion. Since this inverse is cached, it can be reused for evaluating both the gradient GP and the Hessian mean. The training time is identical to a standard GP as the log marginal likelihood objective only operates on the marginal standard GP. Evaluating the mean of the Hessian at a test point scales as $O(N^2 d^2)$. The $d^2$ factor arises from computing all d×d Hessian elements, while $N^2$ comes from the matrix operations with the cached Gram matrix inverse. The optimization subproblem involves solving a second-order cone program (SOCP) with $d+1+m$ decision variables (recall that m is the number of constraints), which has polynomial complexity independent of the number of data points. For our problem scale, CVXOPT provides adequate performance for solving the optimization subproblem, as demonstrated in the runtime results below.
>
> On that note, we want to highlight that we found a more efficient implementation to evaluate the Hessian leveraging Einsum notation vs. iteratively building up the Hessian. With this, we were able to significantly reduce runtime by eliminating redundant computations and leveraging vectorized tensor operations. The optimized implementation achieves almost a $d^2$ speed up as it avoids nested loops and we can now report runtimes on, e.g., the 96D within-model functions setting of TuRBO (WM: 310sec, OHO: 12 206sec), logEI (WM: 186sec, OHO: 1743 sec), BayeSQP (WM: 7sec, OHO: 110sec) where WM is pure within-model, i.e., perfect model knowledge and OHO is with online hyperparameter optimization. As stated above, hyperparameter optimization is similar for all methods.
>
> ---
> ### References
>
> [1] Kandasamy, Kirthevasan, Jeff Schneider, and Barnabás Póczos. "High dimensional Bayesian optimisation and bandits via additive models." International conference on machine learning. PMLR, 2015.
>
> [2] Eriksson, David, and Martin Jankowiak. "High-dimensional Bayesian optimization with sparse axis-aligned subspaces." Uncertainty in Artificial Intelligence. PMLR, 2021.
>
> [3] Wang, Ziyu, et al. "Bayesian optimization in a billion dimensions via random embeddings." Journal of Artificial Intelligence Research 55 (2016): 361-387.
>
> [4] Papenmeier, Leonard, Luigi Nardi, and Matthias Poloczek. "Increasing the scope as you learn: Adaptive Bayesian optimization in nested subspaces." Advances in Neural Information Processing Systems 35 (2022): 11586-11601.

---

> > ### Comment · Reviewer_Br7S · 2025-08-04
> > **Questions sufficiently addressed**
> >
> > Thank you for directly and concisely addressing my questions. I have read through what you have said and I think your suggested changes / updates make me more confident in the paper. I will raise my score.

---

> ### Comment · Area_Chair_ZPkX · 2025-08-04
>
> Dear Reviewer Br7S, given the authors' response, please raise any remaining questions and/or concerns in a timely fashion so the authors have a chance to reply. Thanks!

---

### Decision · Program_Chairs · 2025-09-17

**Decision:**

Accept (spotlight)

**Comment:**

The paper presents a novel algorithm that combines Sequential Quadratic Programming (SQP) with Bayesian Optimization for sample-efficient black-box optimization problems. The key scientific contribution is a novel formulation of SQP in an uncertainty-aware fashion that leverages second-order Gaussian Process surrogates that jointly model function values, gradients, and Hessians using only zero-order observations. The method constructs uncertainty-aware subproblems formulated as second-order cone programs (SOCPs), incorporating Value-at-Risk for objectives and chance constraints for restrictions. The line search to select the next evaluation point is performed via Thompson sampling. Empirical results demonstrate improved performance against strong TuRBO and logEI baselines on synthetic benchmarks, with particular strength in constrained high-dimensional settings.

Strengths:
- Motivation and practical relevance
- Novelty of the methodological approach
- SOCP formulation is a strong technical contribution
- Natural inclusion of constraints
- Strong optimization performance (esp. In higher dimensions, and in constrained settings)
- Clarity of presentation (writing, exposition, polished figures)

Weaknesses
- Empirical evaluation largely limited to synthetic problems (one real-world problem added in rebuttal)
- Local optimization behavior makes method reliant on initialization
- Scalability to (very-)high-dimensional problems is limited

The primary concerns raised by the reviewers were the omission of other scalable Bayesian optimization baselines and real-world benchmarks, as well as the insufficient discussion of some assumptions and experimental details. In their rebuttal, authors comprehensively addressed these concerns by providing comparisons against SAASBO and MPD as additional baselines, additional ablation studies and runtime comparison, and discussion of complexity and scalability. All reviewers were satisfied with the authors’ responses and there were no major outstanding concerns.

The paper makes a substantial contribution to the area of sample-efficient black-box optimization. The technical contribution is significant - successfully bridging classical second-order optimization with modern Bayesian optimization represents a meaningful advance for the constrained optimization community. The uncertainty-aware SOCP formulation is mathematically elegant, computationally tractable, and naturally allows for the inclusion of constraints. While the evaluation focuses on synthetic problems, the consistent strong performance across multiple high-dimensional settings and the addition of baselines like SAASBO and MPD during rebuttal strengthen the empirical evidence for the value of this method. Finally, the presentation is clear and polished and the key limitations are discussed explicitly.

I believe this is an excellent paper - especially if the authors include the additional baselines and discussions produced during the rebuttal period.